# Cerebellar nuclei excitatory neurons regulate developmental scaling of presynaptic Purkinje cell number and organ growth

**Ryan T Willett[1†], N Sumru Bayin[1†], Andrew S Lee[1,2†], Anjana Krishnamurthy[1,2†], Alexandre Wojcinski[1†], Zhimin Lao[1], Daniel Stephen[1], Alberto Rosello-Diez[1‡], Katherine L Dauber-Decker[1], Grant D Orvis[1], Zhuhao Wu[3], Marc Tessier-Lavigne[3], Alexandra L Joyner[1,2,4]***

[1]Developmental Biology Program, Sloan Kettering Institute, New York, United States; [2]Neuroscience Program, Weill Cornell Graduate School of Medical Sciences, New York, United States; [3]The Laboratory of Brain Development and Repair, The Rockefeller University, New York, United States; [4]Biochemistry, Cell and Molecular Biology Program, Weill Cornell Graduate School of Medical Sciences, New York, United States

**Abstract** For neural systems to function effectively, the numbers of each cell type must be proportioned properly during development. We found that conditional knockout of the mouse homeobox genes *En1* and *En2* in the excitatory cerebellar nuclei neurons (eCN) leads to reduced postnatal growth of the cerebellar cortex. A subset of medial and intermediate eCN are lost in the mutants, with an associated cell non-autonomous loss of their presynaptic partner Purkinje cells by birth leading to proportional scaling down of neuron production in the postnatal cerebellar cortex. Genetic killing of embryonic eCN throughout the cerebellum also leads to loss of Purkinje cells and reduced postnatal growth but throughout the cerebellar cortex. Thus, the eCN play a key role in scaling the size of the cerebellum by influencing the survival of their Purkinje cell partners, which in turn regulate production of granule cells and interneurons via the amount of sonic hedgehog secreted.

**\*For correspondence:**
joynera@mskcc.org

[†]These authors contributed equally to this work

**Present address:** [‡]Australian Regenerative Medicine Institute, Monash University, Clayton, Australia

**Competing interests:** The authors declare that no competing interests exist.

## Introduction

A key aspect of brain development is the production of the numerous cell types in the correct proportions so they can assemble into nascent networks. Each cell type is generated during a particular time period and from a defined pool of proliferating stem cells (*Leto et al., 2016*). The number of cells produced is then scaled down based on survival factors produced by target neurons (neurotrophic theory; *Cowan et al., 1984*; *Levi-Montalcini and Hamburger, 1951*; *Oppenheim, 1991*; *Snider, 1994*). The cerebellum represents a powerful system in which to study the phenomenon of cell number scaling during development, since by birth the specification of neuronal lineages is complete but neurogenesis of several cell types has just begun. During the third trimester and first few months of human life the cerebellum undergoes a rapid expansion from a smooth ovoid anlage into a morphologically complex folded structure (*Altman and Bayer, 1997*; *Rakic and Sidman, 1970*). The folds (lobules) are considered the cerebellar cortex, and they overlay the cerebellar nuclei (CN), two bilaterally symmetric groups of mediolaterally-arrayed nuclei in mice (medial, intermediate and lateral nuclei), that house the main output neurons of the cerebellum (*Sillitoe and Joyner, 2007*). The layered cortex has an outer molecular layer, which contains interneurons and the axons of

granule cells and dendrites of Purkinje cells (PCs) and lies above a single layer of PC somata intermixed with the cell bodies of Bergmann glia. Below this layer is a dense layer of granule cells called the inner granule cell layer (IGL). PC axonal projections establish the only electrophysiological and physically direct connection between the cerebellar cortex and the CN. The excitatory CN neurons (eCN) are the first neurons to be born in the embryonic cerebellum, followed by their presynaptic partners, the PCs and local CN interneurons (*Machold and Fishell, 2005*; *Sudarov et al., 2011*). Postnatal cerebellar growth in the mouse is principally driven by the expansion of two progenitor populations that produce the presynaptic partners of PCs, granule cells and interneurons, as well as astrocytes (*Leto et al., 2016*). Genes regulating differentiation of most cerebellar cell types have been identified, but the mechanisms responsible for coordinating the scaling of their cell numbers is poorly understood, despite the known importance for cell type proportions in cerebellar circuit function.

During mouse cerebellar development, neurogenesis occurs in several germinal compartments: 1) the *Atoh1*-expressing rhombic lip which produces the eCN between embryonic day (E) 9.5-E12.5 and then unipolar brush cells directly (*Machold and Fishell, 2005*; *Sekerková et al., 2004*; *Wang et al., 2005*); 2) the ventricular zone which generates inhibitory PCs by E13.5 and early born interneurons including those that populate the CN (*Leto et al., 2016*; *Leto and Rossi, 2012*; *Sudarov et al., 2011*); 3) a rhombic lip-derived pool of granule cell precursors (GCPs) that forms an external granule cell layer (EGL) surrounding the cerebellum from E15.5 to postnatal day (P) 15 with an outer layer of proliferating cells and inner layer of postmitotic granule cells (*Machold and Fishell, 2005*); and 4) a ventricular zone-derived intermediate progenitor pool that expresses nestin, expands after birth and produces astrocytes, including specialized Bergmann glia, and late born interneurons of the molecular layer (*Cerrato et al., 2018*; *Fleming et al., 2013*; *Parmigiani et al., 2015*). The embryonic PCs form a multi-layer beneath the nascent EGL and project to the CN by E15.5 (*Sillitoe et al., 2009*). Once born, granule cells descend along Bergmann glial fibers to form the IGL, and synapse onto PC dendrites in the overlying molecular layer (*Hatten and Heintz, 1995*; *Sillitoe and Joyner, 2007*). PCs secrete sonic hedgehog (SHH) after E17.5, which drives the massive postnatal neurogenic phase by stimulating proliferation of GCPs and nestin-expressing progenitors (*Corrales et al., 2006*; *Corrales et al., 2004*; *De Luca et al., 2015*; *Fleming et al., 2013*; *Lewis et al., 2004*; *Parmigiani et al., 2015*; *Wojcinski et al., 2017*). PCs thus act in a dual role as a synaptic bridge between the two main rhombic lip-derived glutamatergic neuronal subtypes and as a developmental regulator of postnatal cortex growth driven by proliferation. Scaling of the proportions of postnatally born cells in the cerebellar cortex has been proposed to be regulated by the number of PCs and the amount of SHH they produce (*Fleming and Chiang, 2015*; *Fleming et al., 2013*). However, how the number of eCN neurons and PCs are proportioned during embryogenesis has not been addressed. Furthermore, it is not known whether PCs in the cerebellar cortex are dependent on eCN neurons for their development or scaling.

Engrailed 1 and 2 (*En1/2*) genes encoding homeobox transcription factors provide a powerful genetic entry point for studying developmental scaling of neuron types, since *En1/2* conditional or null mutants have a seemingly well-preserved cytoarchitecture despite suffering cerebellar hypoplasia that preferentially affects particular lobules (*Cheng et al., 2010*; *Millen et al., 1994*; *Orvis et al., 2012*; *Sgaier et al., 2005*). For example, specific loss of *En1/2* in the *Atoh1*-expressing rhombic lip-lineage (*Atoh1-Cre; En1^{lox/lox}; En2^{lox/lox}* conditional knockouts, referred to as *Atoh1-En1/2* CKOs) results in preferential loss of cerebellum volume in the medial cerebellum (vermis and paravermis), with anterior/central region foliation defects (*Figure 1A*; *Orvis et al., 2012*). As a basis for studying the roles of the *En1/2* genes in scaling of cerebellar neurons, we confirmed that the numbers of GCs, PCs, and molecular layer interneurons in the mutants are scaled down in numbers relative to the decrease in cerebellar area, while largely preserving their densities. *Atoh1-En1/2* CKOs nevertheless have motor behavior deficits. The first defect in *Atoh1-En1/2* CKOs was discovered to be death of a subset of eCN neurons after E14.5 in the medial and intermediate nuclei. The early loss of eCN is accompanied by a cell non-autonomous loss of PCs in *Atoh1-En1/2* CKOs. Deletion of *En1/2* in the cerebellum only in GCPs or eCN (*GCP-En1/2* or *eCN-En1/2* CKOs) revealed that *En1/2* play only a minor role in promoting differentiation of GCPs but a major role in viability of a subset of medial and intermediate eCN and secondarily in differential survival of PCs and corresponding cortex growth in the anterior and central regions of the vermis and paravermis. Circuit mapping further revealed that the PCs in the anterior or central regions of the vermis project to different regions of

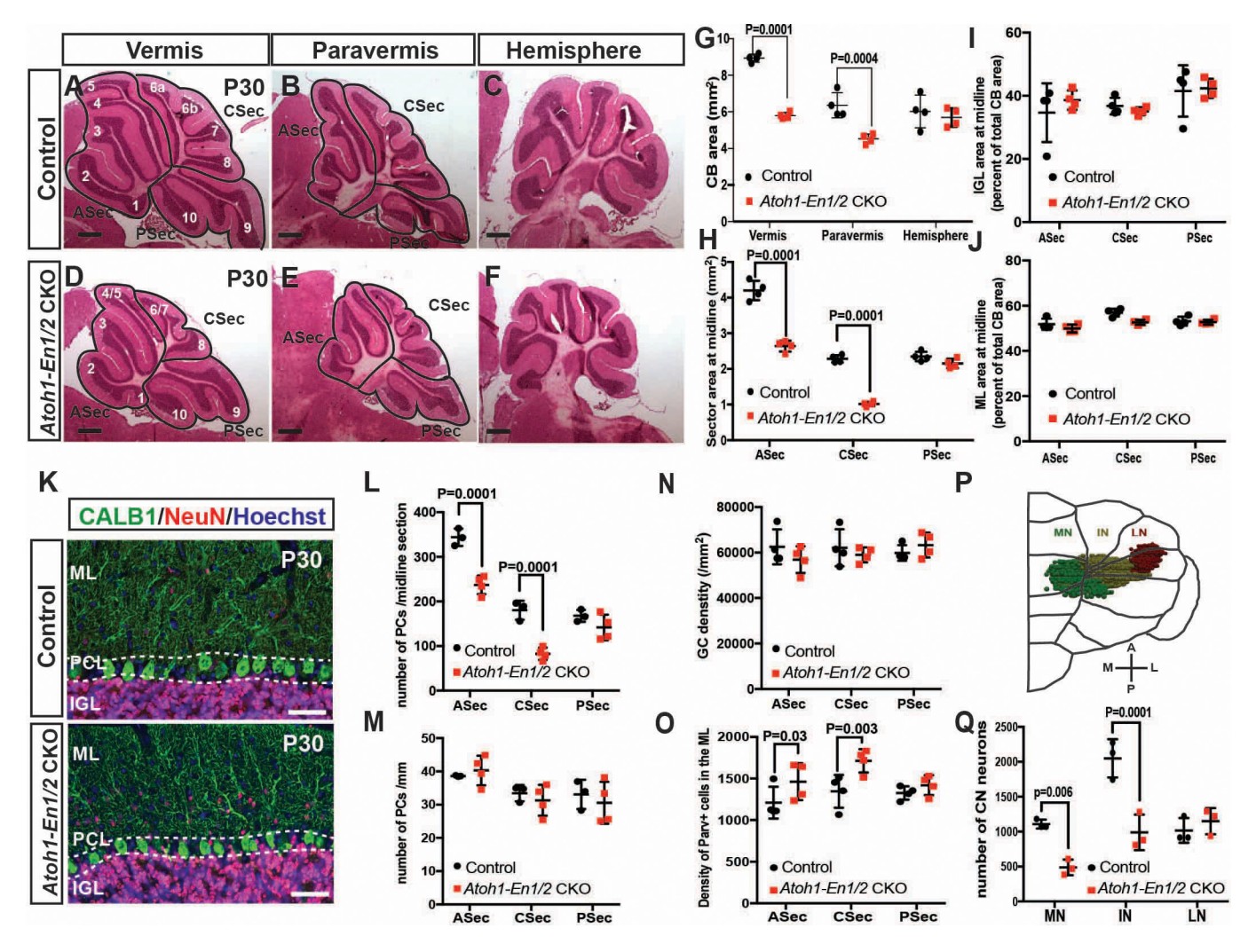

**Figure 1.** Loss of *En1/2* in the rhombic lip-lineage results in reduced growth of the anterior and central vermis and paravermis with scaling of neuron numbers. (**A-F**) H and E staining of sagittal sections from the midline (vermis), paravermis and hemispheres of P30 mutant and control cerebella showing reduction of the anterior and central sectors (ASec and CSec) and not the posterior sector (PSec) specifically in the vermis and paravermis. (**G**) Quantification of the total cerebellum area in the vermis, paravermis and hemisphere (n = 4 animals/condition, Two-way ANOVA, $F_{(1,6)}$=43.14, p<0.0006). (**H**) Quantification of sector areas in the vermis of P30 control and *Atoh1-En1/2* CKO animals (n = 4 animals/condition, Two-way ANOVA, $F_{(1,9)}$=398.277, p<0.0001). (**I–J**) IGL (**I**) and molecular layer (**J**) sector area quantifications in the vermis as the percent of total average area showing no change in *Atoh1-En1/2* CKOs compared to controls (n = 4 animals/condition). **K**) Immunofluorescence analysis of P30 cerebellar sections for the PC marker Calbindin1 (CALB1) and the pan-neuronal marker NeuN in a *Atoh-En1/2* CKO (**G**) compared to a control. (**L–M**) Quantification of average PC numbers in each sector per midline sagittal section (**L**) showing reductions only in the ASec and CSec, whereas the density of PCs (**M**) is conserved (n = 3 for controls and n = 4 for *Atoh1-En1/2* CKO, J: Two-way ANOVA, $F_{(1,15)}$=72.52, p<0.0001). (**N**) Quantification of granule cell density in each vermal sector of mutants and controls (n = 4 for each genotype). **O**) Quantification of the density of ParV+ cells in the ML per sector of mutants compared to controls (n = 4 for each genotype, Two-way ANOVA, $F_{(1,9)}$=28.4, p<0.0005). (**P**) Schematic representation of a half brain with a 3D reconstruction of the eCN in a normal cerebellum. (**Q**) Quantification of eCN neurons in the medial (MN), intermediate (IN) and lateral nuclei (LN) of the CN in one half of *Atoh1-En1/2* CKO cerebella compared to littermate controls (n = 3 per genotype) (Two-way ANOVA, $F_{(1,12)}$=32.29, p=0.0001). Significant *post hoc* comparisons are shown in the figure. CKO: conditional knockout, IGL: internal granule layer, ML: molecular layer, PCL: Purkinje cell layer, PC: Purkinje cell, GC: granule cell. Scale bars B: 500 μm, F-I: 100 μm. Lobule numbers are designated in *Figure 1A*.
The online version of this article includes the following source data and figure supplement(s) for figure 1:

**Source data 1.** Summary of the statistics.
**Figure supplement 1.** *Atoh1-En1/2* CKOs have reduced area and PC numbers in the ASec and PSec in the vermis and paravermis but the densities and ratios of neurons remain similar to normal.
**Figure supplement 2.** Proportion of EGL that contains proliferating GCPs to total EGL area is preserved in *Atoh1-En1/2* CKO animals.
**Figure supplement 3.** 3D reconstructions of stereology of the eCN in *Atoh1-En1/2* CKOs and controls.

the medial CN (anterior and posterior, respectively). The region-specific scaling of the cerebellar cortex thus could depend on the degree to which particular eCN subpopulations are reduced in the *Atoh1-En1/2* CKOs. Demonstrating that PC numbers are reduced in number when eCN are reduced, we showed that when ~ 40% of embryonic eCN are genetically killed using attenuated diphtheria toxin (DTA) in all three nuclei, PC numbers and cortex growth are correspondingly reduced throughout the cerebellum. We propose a model whereby the number of eCN neurons is involved in setting the growth potential of the cerebellar cortex through supporting survival of a balanced population of PCs that then stimulate proliferation of granule cell and interneuron progenitors.

## Results

### Loss of *En1/2* in the rhombic lip-lineage results in reduced growth of the anterior and central vermis and paravermis with scaling of neuron numbers

Our previous study using 3D Magnetic Resonance Imaging showed a preferential reduction (25%) in the volume of the medial cerebellum (vermis and paravermis) of *Atoh1-En1/2* CKOs (*Orvis et al., 2012*). Although the cytoarchitecture of *Atoh1-En1/2* CKOs appeared preserved, whether scaling of neuron numbers was intact in the mutants was not tested. We therefore determined whether the proportions of different neuron types was preserved in areas of the cerebellum with reduced size. As a proxy for analyzing cerebellum size, we quantified the area of mid-sagittal, paravermis and hemisphere sections of ~P30 animals, and found a $31.2 \pm 2.0\%$ and $28.8 \pm 4.1\%$ reduction in the vermis and paravermis, respectively but no reduction in the hemispheres of *Atoh1-En1/2* CKOs compared to littermate controls (*En1^{lox/lox}; En2^{lox/lox}* mice) (*Figure 1A–G*). When the vermis and paravermis were divided into three regions - an anterior sector (ASec), anterior to the primary fissure (lobules 1–5); a central sector (CSec), between the primary and secondary fissures (lobules 6–8); and a posterior sector (PSec), posterior to the secondary fissure (lobules 9–10), the PSec had no significant reduction, whereas the ASec and CSec were significantly decreased in area in *Atoh1-En1/2* CKOs compared to littermate controls (ASec and CSec of the midline: $37.1 \pm 3.6\%$ and $55.4 \pm 2.7\%$; paravermis: $25.0 \pm 1.4\%$ and $46.7 \pm 7.0\%$) (*Figure 1H*, *Figure 1—figure supplement 1A*). Thus, *Atoh1-En1/2* CKOs have a specific and differential reduction in the area of the anterior and central regions of the vermis and paravermis.

We next determined whether the areas of the cell layers in the sectors of the vermis and paravermis (normalized to the total areas of each sector) were scaled proportionally in *Atoh1-En1/2* CKOs. As predicted, in *Atoh1-En1/2* CKOs the areas of the IGL and molecular layer were reduced only in the ASec and CSec of the vermis and paravermis and their proportions were well conserved compared to controls (*Figure 1I,J* and *Figure 1—figure supplement 1B–E,G,H*). In order to determine whether the numbers of each cell type in the sectors of the cerebellar cortex were scaled down, we quantified the number and the density of the three major neuron types. The numbers of CALB1+ PCs per midline, paravermis or hemisphere section at ~P30 were indeed found to be reduced specifically in the ASec and CSec of the vermis ($31.0 \pm 6.0\%$ and $54.1 \pm 7.5\%$ reduction, respectively) and paravermis ($30.9 \pm 7.9\%$ and $37.7 \pm 11\%$ reduction, respectively), but their densities remained normal in all sectors of *Atoh1-En1/2* CKOs compared to controls (*Figure 1K–M*, *Figure 1—figure supplement 1F,I*), indicating their numbers had scaled down proportionally. Similarly, quantification of NeuN-expressing granule cells showed that their densities were normal in all three sectors of the vermis (*Figure 1N*). Furthermore, the ratio of granule cells to PCs was preserved in the vermis (*Figure 1—figure supplement 1J*). The ratio of interneurons to PCs or granule cells was preserved in the three sectors of *Atoh1-En1/2* CKOs compared to controls (*Figure 1—figure supplement 1K,L*), although the density of Parvalbumin+ molecular layer interneurons was mildly increased in the ASec and CSec of mutants ($1.2 \pm 0.2$ and $1.3 \pm 0.1$ fold, respectively, *Figure 1O*). Thus, the major neuron types in the cerebellar cortex are scaled down regionally in numbers and the cell-to-cell ratios of the cortical neurons of the cerebellum are well preserved. Indeed, quantification of the proportion of the EGL that contains the outer layer of proliferating GCPs at P7 showed that the ratio was preserved despite the EGL length being reduced in the ASec and CSec (*Figure 1—figure supplement 2*).

## *Atoh1-En1/2* mutants have motor deficits

We next tested whether a smaller cerebellum that has a well scaled cortex can support normal motor behavior, since we had found that irradiated mice with an ~20% reduction in the area of mid-sagittal sections but well scaled layers have no obvious motor defects (*Wojcinski et al., 2017*). Two motor behavior paradigms and grip strength were tested. Unlike irradiated mice, significant defects in motor behavior were observed in two cohorts of *Atoh1-En1/2* CKOs (1-month-old and 3-month-old). The performance of mutants on trials of the accelerating rotarod over time showed a significant difference, as well as the cumulative measurement of performance in *Atoh1-En1/2* CKOs, with a shorter

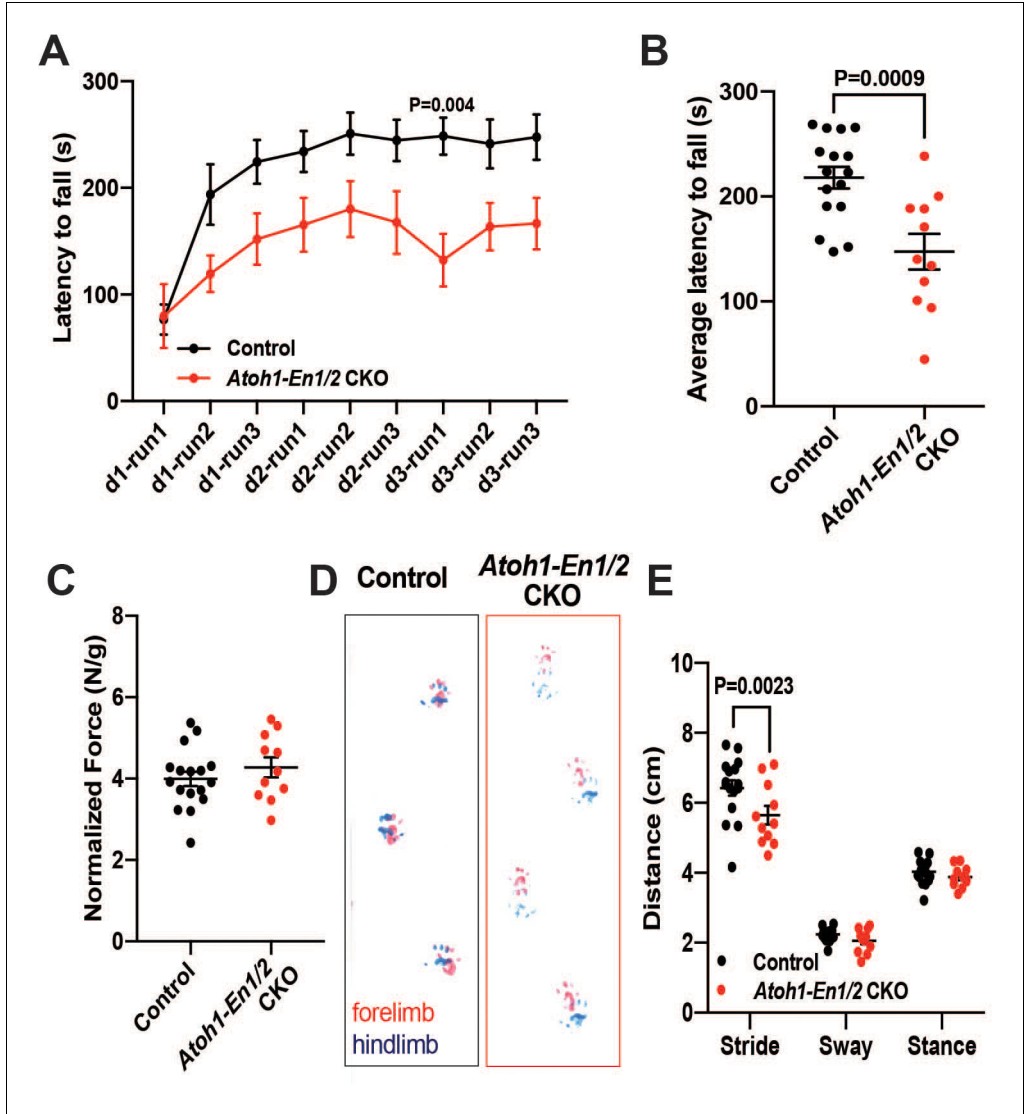

**Figure 2.** *Atoh1-En1/2* CKO animals show motor behavior defects compared to control animals. (A) Latency to fall from rotarod at each trial (Two-way ANOVA, $F_{(1,25)}$=14.23, p=0.0009). (B) Average latency to fall for 12 week old control animals (n = 17) compared to *Atoh1-En1/2* CKOs (n = 11) (Student's t-test). (C) Quantification of grip strength analysis of *Atoh1-En1/2* CKO animals (n = 11) compared to controls (n = 17) (Student's t-test). (D–E) Representative images (D) and quantification (E) of footprint analysis performed on control (n = 17) and *Atoh1-En1/2* CKO (or *En1/2* CKO in B) (n = 11) animals (Two-way ANOVA, $F_{(1,75)}$=8.227, p=0.005). Significant *post hoc* comparisons are shown in the figure.

The online version of this article includes the following figure supplement(s) for figure 2:

**Figure supplement 1.** *Atoh1-En1/2* CKO animals show motor behavior defects compared to control animals at P30.

latency to fall compared to littermate controls at both ages (*Figure 2A–B* and *Figure 2—figure supplement 1A–B*). *Atoh1-En1/2* CKOs did not have a significant decrease in grip strength compared to littermate controls at 3 months (*Figure 2C*) although they did at 1 month (*Figure 2—figure supplement 1D*). In a footprint analysis of gait, mutant mice had a significant decrease in stride length at both ages (*Figure 2D–E* and *Figure 2—figure supplement 1B-C*). Thus, *Atoh1-En1/2* CKOs have motor behavior deficits despite the numbers of cells in the cerebellar cortical circuit being well scaled.

## Loss of *En1/2* in the rhombic lip-lineage results in a preferential loss of medial and intermediate eCN

Given the motor deficits, we asked whether the number of rhombic lip-derived eCN are altered in *Atoh1-En1/2* CKOs. Stereology of Nissl-stained large neurons on every other coronal section of half cerebella of *Atoh1-En1/2* CKOs and littermates revealed an ~37.5 ± 4.6% reduction in the total number of large neurons in the *Atoh1-En1/2* CKOs compared to controls. Moreover, the loss of large neurons was specific to the medial and intermediate nuclei (*Figure 1Q*, *Figure 1—figure supplement 3* and *Videos 1–2*). Since PCs in the vermis and hemispheres project to the medial and intermediate CN (*Leto et al., 2016*), there is a correlation between the eCN that are reduced in number in *Atoh1-En1/2* CKOs and the areas of the cerebellum with a growth deficit (paravermis and vermis) in terms of the PC projection maps. We next assessed whether scaling of PCs and eCN was preserved in adult mutants. Since the PCs in lobules 1–8 project primarily to the medial CN whereas most of lobule nine and all of 10 project to the vestibular nucleus (*Leto et al., 2016*; *Walberg and Dietrichs, 1988*), we compared the decrease in the number of eCN in the medial nucleus of *Atoh1-En1/2* CKOs to the number of PCs in the ASec and CSec of a half vermis. Interestingly, there was an estimated 1.4 ± 0.1 fold increase in the ratio of the number of PCs to eCN in *Atoh1-En1/2* CKOs compared to controls (*Figure 1—figure supplement 1M*). Thus, it appears that the PCs numbers did not fully scale down in proportion to the reduction in medial eCN.

## ASec and CSec PCs project to distinct regions of the medial CN

Given that PCs are known to broadly innervate the nuclei closest to them (*Leto et al., 2016*), one interpretation of our results is that the presynaptic PC partners of the eCN that are depleted in *Atoh1-En1/2* CKOs are the ones that are preferentially lost in mutants, since the hemispheres are spared as well as the posterior lobules of the medial cerebellum. Quantification of the medial CN that remained in *Atoh1-En1/2* mutants compared to controls indicated that there is a preferential loss of the posterior part of the nucleus (see *Videos 1* and *2* and *Figure 1—figure supplement 3*). Although a comprehensive study of topology based on the anterior-posterior position of PCs in lobules has not been published, it has been shown that neurons of the more posterior region of the medial CN (fastigial oculomotor region) receive functional inputs from PCs in lobules 6 and 7 of the vermis (*Herzfeld et al., 2015*; *Noda et al., 1990*; *Sugihara et al., 2009*; *Zhang et al., 2016*). We therefore asked whether PCs in the ASec and CSec project to different regions of the medial CN by injecting a Cre-inducible tracer virus (AAV5-EF1a-DIO-mCherry) into lobules 3–5 or lobules 6/7 of ~P30 *Pcp2^{Cre/+}* mice (*Figure 3A–B*) that express Cre recombinase in PCs (*Zhang et al., 2004*). 1.5 weeks post-injection, brains underwent delipidation to enhance immunofluorescence in the white matter (*Renier et al., 2014*) and free-floating 50 µm-thick coronal sections were immunostained. Interestingly, we found that labeled PCs in the ASec preferentially innervated the anterior medial CN (*Figure 3C–F*), whereas labeled PCs

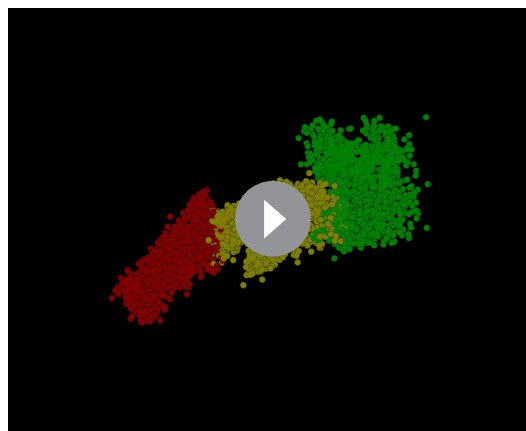

**Video 1.** Stereology 3D reconstruction of a half cerebellum from a control animal at P30. Green: medial CN, yellow: Intermediate CM, red: Lateral CN.
https://elifesciences.org/articles/50617#video1

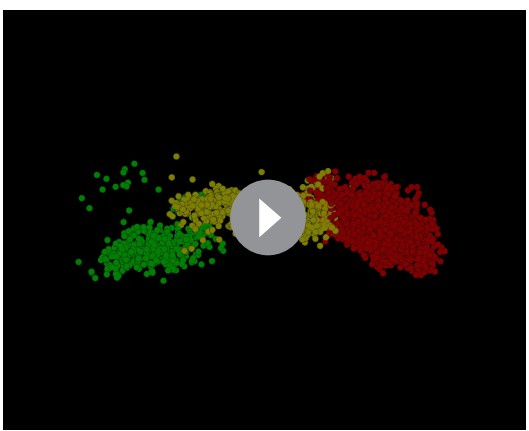

**Video 2.** Stereology 3D reconstruction of a half cerebellum from a *Atoh-EN1/2* CKO animal at P30. Green: medial CN, yellow: Intermediate CM, red: Lateral CN.

https://elifesciences.org/articles/50617#video2

from the CSec, the region with the greatest growth defect, innervated the posterior medial CN (*Figure 3C,G–I*). Some labeled PCs in the vermis also targeted the intermediate CN (*Figure 3F and G*). These results provide evidence that there could be a causal relationship between the extent to which the eCN are reduced in number in particular locations within each CN and the reduction in the number of their specific presynaptic PCs and associated lobule growth.

## *En1/2* are transiently expressed in the rhombic lip-lineage during embryonic development

We previously showed that *En1/2* are expressed in subsets of all cerebellar cell types from as early as E8.5 (*Davis et al., 1991*; *Millen et al., 1995*; *Wilson et al., 2011*) including in a small percentage of the medial TBR1+ eCN at E17.5 and granule cells primarily after they stop proliferating (*Wilson et al., 2011*). In order to identify which neurons could be responsible for the *Atoh1-En1/2* CKO phenotype, we determined which cells undergo *Atoh1-Cre* recombination and when EN1/2 are expressed in the cells.

First, we confirmed that *Atoh1-Cre* recombines in most eCN by generating mice carrying *Atoh1-Cre* and the *R26^{LSL-nTDTom}* reporter allele (*Atoh1-nTDTom* mice) and co-labeled for nuclear TDTom and three eCN markers (NeuN, MEIS2 protein and *Slc17a6* RNA). All but scattered TDTom+ cells expressed NeuN and the vast majority expressed *Slc17a6* (vGlut2), and very few cells that expressed either marker did not express TDTom (*Figure 4—figure supplement 1A–R*). Almost all TDTom+ cells also expressed MEIS2, although at intermediate levels along the medial-lateral axis the co-labeling was less extensive (*Figure 4—figure supplement 1M–O*). Furthermore, almost all cells expressing *Slc6a5* RNA (marker for glycinergic cells) or *Gad1* (marker for inhibitory neurons) were negative for TDTom (*Figure 4—figure supplement 1S,T*). As expected, all granule cells in the IGL of the lobules anterior to lobule nine were labeled, with only partial labeling in lobules 9 and 10, and no labeling of PCs or interneurons (*Figure 4—figure supplement 1P*).

We then examined expression of EN1/2 in the developing rhombic lip-lineage in more detail using an antibody that detects both proteins (*Davis et al., 1991*) and a new line of mice in which EN2 protein is tagged at the C-terminus with three HA epitopes (*En2^{HA/HA}*, see Materials and methods). Using *Atoh1-TDTom* mice, we found that at E12.5, EN1/2 are expressed in the TDTom+ cells of the forming nuclear transitory zone that houses the newly born eCN and the PCs that have migrated away from the ventricular zone (*Figure 4A,B*, *Figure 4—figure supplement 2A–C*). EN1/2 expression levels varied in eCN subpopulations as development progressed. At E15.5, high levels of EN1/2 were observed in TDTom+ cells in the medial and intermediate CN, whereas lateral CN showed reduced expression in TDTom+ cells (*Figure 4C,D*). GCPs in the EGL and PCs expressed EN1/2 at high levels at E15.5 (*Figure 4E*, *Figure 4—figure supplement 2D–F*). EN1/2 expression in eCN was confirmed using *En2^{HA/HA}* animals (*Figure 4I–P*). Immunofluorescent analysis of adjacent sections of E15.5 embryos with MEIS2 and HA antibodies showed expression of EN2 in eCN. Interestingly, only rare eCN expressed EN1 and not EN2 (*Figure 4I–P*). By E17.5, the majority of eCN cells had downregulated EN1/2, however scattered populations of TDTom+ cells continued to maintain strong expression, likely in the intermediate nuclei (*Figure 4F–G*). At E17.5 EN1/2 were strongly expressed in the inner EGL, where postmitotic GCs reside (*Figure 4H*). Finally, in E13.5 *Atoh1-En1/2* CKOs there was not yet a major decrease in EN1/2 protein in the eCN (*Figure 4—figure supplement 3*). However, complete loss of EN1/2 protein in the eCN and EGL was observed by E15.5, whereas as expected EN1/2 expression in the PCs did not appear altered (*Figure 4Q–X*). Overall, this data shows that EN1/2 are broadly expressed in both rhombic lip- and ventricular zone-derived

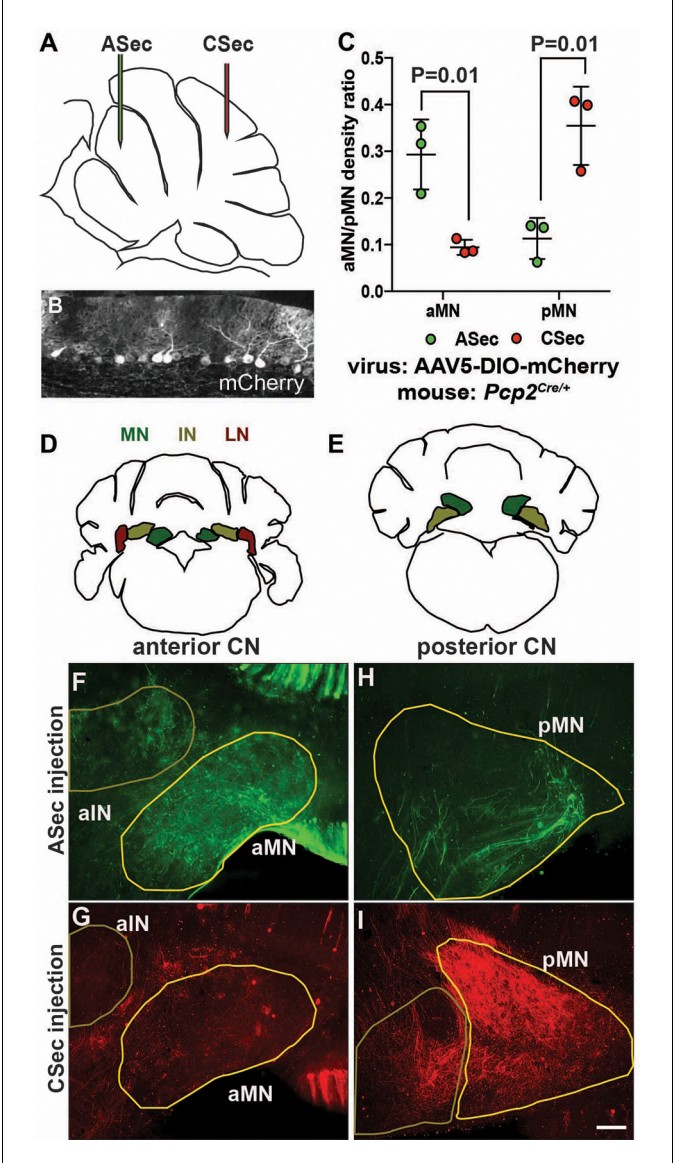

**Figure 3.** The anterior and central sectors of the vermis preferentially innervate different subregions of the medial CN. (A) Diagram of scheme used for stereotaxic injection of AAV-DIO-mCherry reporter virus to label PC axons projecting to the CN from the ASec or CSec of *Pcp2*$^{Cre/+}$ animals. (B) Sagittal section the PC layer of a *Pcp2*$^{Cre/+}$ mouse injected with AAV-DIO-mCherry showing successful targeting of PCs. (C) Quantification of fluorescence signal density in the anterior medial nucleus (aMN) and posterior medial nucleus (pMN) in animals injected with virus in the ASec or CSec showing that the ASec PCs preferentially innervate the aMN (n = 3, Student's t-test) and CSec labeled PCs preferentially innervate the pMN (n = 3, Student's t-test). (D–E) Schematic representations of the anterior (D) and posterior (E) CN. (F–I). Immunofluorescent analysis showing the preferential labeling of the aMN or pMN in mice injected to trace axons projecting from PCs in the ASec (F–G) or CSec (H–I), respectively. ASec injections and CSec are pseudo-colored in green and red, respectively. Scale bars: 150 μm.

cells of the cerebellum, however their expression is spatially and temporally dynamic and EN1/2 expression is not lost in the eCN until E15.5 in mutants.

## *En1/2* are required in the eCN but not GCPs for growth of the cerebellar cortex

The cerebellar hypoplasia seen in *Atoh1-En1/2* CKOs could be due to a cell autonomous requirement for *En1/2* in eCN and/or in GCPs. In GCPs *En1/2* could promote their expansion. Alternatively,

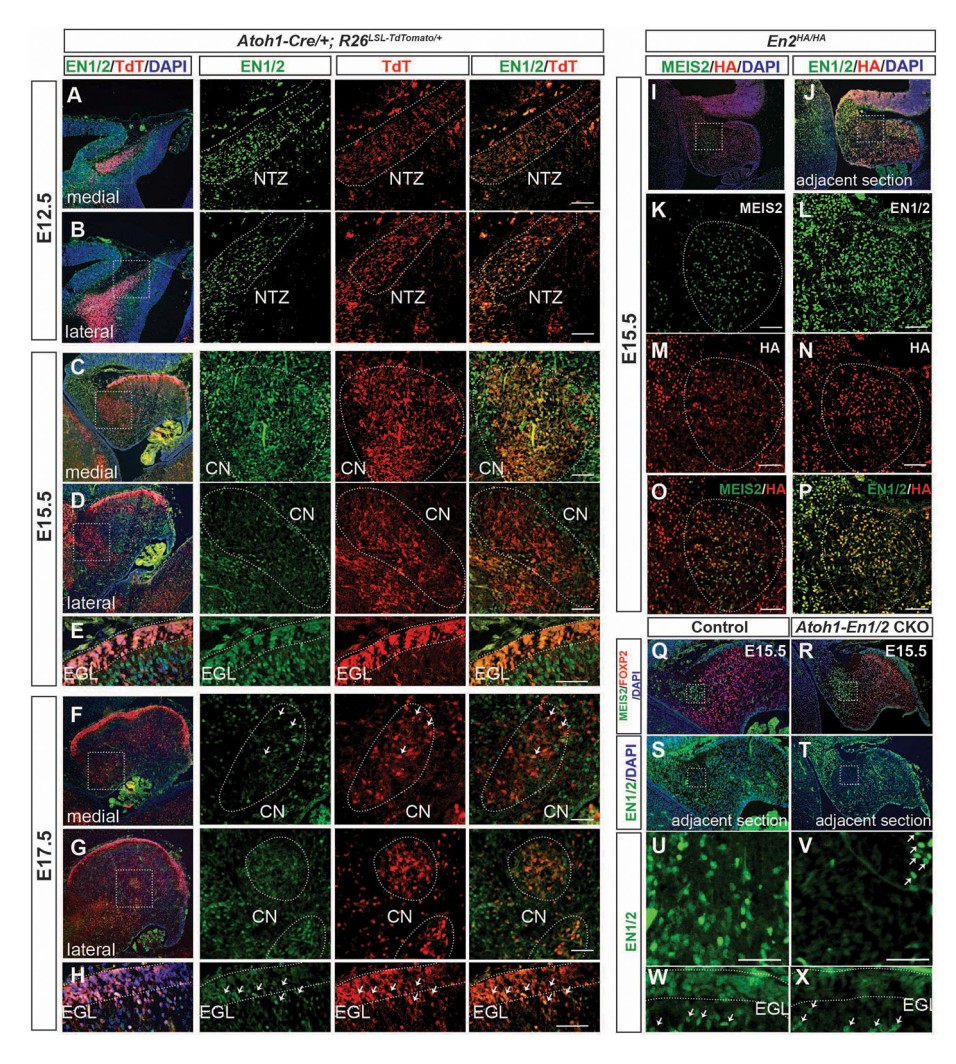

**Figure 4.** EN1/2 are dynamically expressed in the rhombic lip-lineage during embryonic development. (**A–B**) Immunofluorescent analysis using a pan-EN antibody on sections of E12.5 *Atoh1-Cre/+; R26*$^{LSL-TdTom/+}$ embryos showing EN1/2 expression in the TdTomato-expressing (TdT+) cells in the nuclear transitory zone (NTZ) through the mediolateral axis of the developing cerebellum. (**C–D**) Immunofluorescent analysis of E15.5 sections showing EN1/2 expression is mainly in a subset of TdT+ eCN in the medial cerebellum, with scattered expression in lateral TdT+ cells. (**E**) Immunofluorescent analysis of EN1/2 expression in the EGL at E15.5. (**F–G**) Immunofluorescent analysis showing that by E17.5 the majority of the TdT+ cells in the CN no longer express EN1/2 except for a subset of eCN (arrows) in the medial/intermediate eCN. (**H**) Immunofluorescent analysis of the EGL at E17.5 showing higher nuclear expression of EN1/2 in the inner EGL (arrows). (**I–P**) EN1/2 expression was confirmed using an *En2*$^{HA/HA}$ mouse line expressing 3xHA-tagged EN2 protein at E15.5 and co-staining sections with pan-EN and anti-HA antibodies. The eCN area was identified using immunofluorescent analysis of adjacent sections for MEIS2. (**Q–X**) Immunofluorescent analysis of sections from E15.5 *Atoh1-En1/2* CKOs and their littermate controls showing loss of EN1/2 expression in the eCN (insets: U,V) and EGL (**W,X**). EN1/2 expression in the PCs is not affected as expected (arrows). MEIS2 and FOXP2 staining of adjacent sections were used to identify eCN and PCs. EGL outlines were determined based on DAPI staining. Scale bars: 100 μm.

The online version of this article includes the following figure supplement(s) for figure 4:

**Figure supplement 1.** *Atoh1-Cre* marked cells expressing TDTom also express eCN markers (MEIS2 and *Slc17a6*) and a pan neuronal marker (NeuN) but not a glycinergic (*Slc6a5*) or GABAergic (*Gad1*) neuron marker.

**Figure supplement 2.** Dynamic embryonic expression of EN1/2 in PCs during development.

**Figure supplement 3.** Analysis of EN1/2 expression at E13.5 in *Atoh1-TDTom-En1/2* CKOs and their littermate control embryos.

the reduced number of PCs could be a consequence of the loss of eCN and therefore a decrease in SHH-stimulated proliferation of GCPs and nestin-expressing progenitors that give rise to interneurons (*Corrales et al., 2006*; *De Luca et al., 2015*; *Fleming et al., 2013*; *Lewis et al., 2004*; *Wojcinski et al., 2017*). As a means to resolve this question, we mutated *En1/2* separately in GCPs or eCN in the cerebellum by taking advantage of the different neurogenic periods for the two cell types in the rhombic lip-lineage (*Machold and Fishell, 2005*). Using a new *Atoh1-tTA* transgenic line (see Materials and methods) and a *tetO-Cre* line (*Perl et al., 2002*) we established two doxycycline (Dox) regimens that result in embryonic Cre-mediated recombination in the cerebellum primarily in

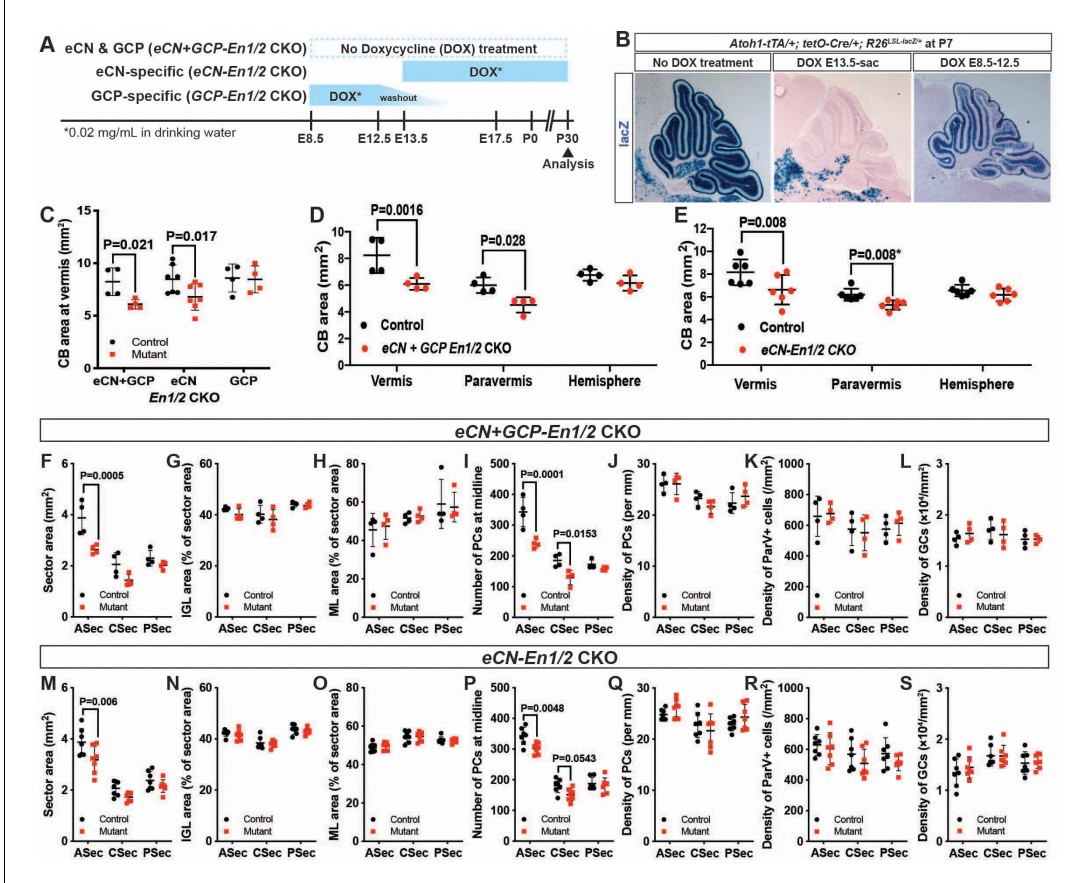

**Figure 5.** *En1/2* are required in the eCN but not GCPs for growth of the cerebellar cortex. (A) Schematic explaining genetics used to delete *En1/2* in the eCN and/or GCPs. *Atoh1-tTA; tetO-Cre; En1^{lox/lox}; En2^{lox/lox}* animals were treated with Dox from either E13.5 until sacrifice to specifically knockout *En1/2* in eCN (*eCN-En1/2* CKO) or between E8.5 and E12.5 to knockout *En1/2* in GCPs (*GCP-En1/2* CKO). No Dox was given to knockout *En1/2* in both cell types in the cerebellum (*eCN+GCP-En1/2* CKO). (B) Staining for lacZ activity on sections of *Atoh1-tTA; tetO-Cre; R26^{LSL-LacZ/+}* animals at P7 showing the specificity of the Dox regimens. (C) Quantification of cerebellum area in the vermis showing reduction only in *eCN+GCP-* and *eCN-En1/2* CKOs compared to their littermate controls at P30 (n = 4 animals/condition, Two-way ANOVA, $F_{(1,24)}=8.042$, p=0.009). (D,E) Quantification of cerebellum area based on location showing the area is only reduced in the vermis and paravermis, but not in the hemispheres at P30 in *eCN+GCP-En1/2* CKOs (D, n = 4 animals/condition Two-way ANOVA, $F_{(1,6)}=12.88$, p=0.011) and *eCN-En1/2* CKOs (E, n = 6 animals/condition, Two-way ANOVA, $F_{(1,10)}=8.34$, p=0.016, *Student's t-test in the paravermis). (F–S) Quantifications by sector in the vermis for: overall area (F,M), for IGL (G,N) or ML (H, O) area relative to total sector area, for PC numbers (I,P) and for density of PCs (J,Q), or ML interneurons (K,R) or GCs (L, S) in *eCN+GCP-En1/2* CKOs and *eCN-En1/2* CKOs. A reduction in the ASec area is seen in both mutants (*eCN+GCP-En1/2* CKO: Two-way ANOVA, $F_{(1,18)}=22.07$, p=0.0002, *eCN-En1/2* CKOs: Two-way ANOVA, $F_{(1,36)}=12.4$, p=0.0012) and PC numbers in the ASec (*eCN+GCP-En1/2* CKO: Two-way ANOVA, $F_{(1,18)}=32.51$, p<0.0001, *eCN-En1/2* CKOs: Two-way ANOVA, $F_{(1,36)}=13.5$, p=0.0008), however, cell densities are preserved. Significant *post hoc* comparisons are shown in the figure. The online version of this article includes the following figure supplement(s) for figure 5:

**Figure supplement 1.** Cell-type specific fate mapping in *Atoh1-tTA; tetO-Cre; R26^{LSL-nTDTom/+}* mice.

**Figure supplement 2.** *En1/2* CKOs involving the eCN have reduced area and PC numbers in the paravermis but the densities and ratios of neurons remain similar to normal.

**Figure supplement 3.** *GCP-En1/2* CKOs do not show a major cerebellar growth defect and *En1/2* promote differentiation of granule cells.

the eCN or the GCPs/granule cells (*Figure 5A*). During administration of Dox, Cre recombination is blocked and then within 3–4 days of Dox withdrawal Cre recombination is restored (*Waclaw et al., 2009*; *Perl et al., 2002*). Given that eCN are generated from *Atoh1*-expressing cells between E9.5-E12.5 (*Machold and Fishell, 2005*), we found that treatment of animals with Dox from E13.5 until sacrifice resulted in recombination in the cerebellum primarily in the eCN of *Atoh1-tTA/+; tetO-Cre/+; R26^{LSL-lacZ/+}* or *Atoh1-tTA/+; tetO-Cre/+; R26^{LSL-nTDTom/+}* mice, whereas Dox administration from E8.5 to E12.5 lead primarily to targeting the GCPs and GCs (*Figure 5B*, *Figure 5—figure supplement 1A–L*). With no Dox treatment, both eCN, GCPs and GCs were targeted (lacZ+, *Figure 5B* or nTDTom+, *Figure 5—figure supplement 1A–D*). In addition, consistent with when most precerebellar nuclei are born, many of the nuclei were lacZ+ in *Atoh1-tTA/+; tetO-Cre/+; R26^{LSL-lacZ/+}* mice (*Figure 5B*). Unexpectedly, the *tetO-Cre* line alone results in nonspecific recombination in neurons in the forebrain and in a small number of PCs in the cerebellum mainly after P7 (*Figure 5—figure supplement 1M–O*).

When the pharmacogenetic strategies were used to ablate *En1/2* in *Atoh1-tTA; tetO-Cre; En1^{lox/lox}; En2^{lox/lox}* mice, we found that mutation of *En1/2* specifically in GCPs (*GCP-En1/2* CKO) did not lead to a reduction in cerebellar growth compared to littermate controls (*Figure 5C*). Significantly, deletion of *En1/2* in embryonic eCN+GCPs (no Dox) or eCN alone resulted in a reduction in the area of the vermis and the paravermis but not the hemispheres compared to littermate controls (*Figure 5D–E*), with the significant decrease being in the ASec (*Figure 5F,M*, *Figure 5—figure supplement 2A,G*). The areas of the IGL and molecular layer were reduced in the ASec of the vermis and paravermis and their proportions were conserved relative to the total area of each sector in both *eCN+GCP-En1/2* and *eCN-En1/2* CKOs (*Figure 5G,H,N,O* and *Figure 5—figure supplement 2D,E,J,K*). Similarly, the number of PCs were reduced in the ASec of the vermis and paravermis in both *eCN+GCP-En1/2* and *eCN-En1/2* CKOs, but PC densities were unchanged (*Figure 5I,J,P,Q* and *Figure 5—figure supplement 2B,C,H,I*). We also did not observe any significant changes in the densities of GCs (*Figure 5L,S*) and molecular layer interneurons (Pvalb+) (*Figure 5K,R*). Finally, the ratio of GCs or INs to PCs was conserved in both mutants (*Figure 5—figure supplement 2F,L*). Taken together, our data demonstrate that *En1/2* are necessary in the eCN compartment of the rhombic lip-lineage to promote postnatal growth of the cerebellar cortex. The numbers of interneurons and granule cells in the cerebellar cortex are scaled down proportionately to the number of PCs, likely through decreased SHH-stimulated proliferation of progenitors in *eCN-En1/2* CKOs as in *Atoh1-En1/2* CKO (*Corrales et al., 2006*; *De Luca et al., 2015*; *Fleming et al., 2013*; *Lewis et al., 2004*; *Wojcinski et al., 2017*).

Given the smaller cerebellum in *eCN+GCP-En1/2* and *eCN-En1/2* CKOs we determined the number of eCN using NeuN immunohistochemistry (*Figure 6A*, *Figure 6—figure supplement 1*). Quantification of the number of large NeuN+ cells (cell area = 100–600 um$^2$) on every other sagittal section of half of the ~P30 cerebellum (see Material and methods, *Figure 6—figure supplement 1C*) showed a 28.2 ± 2.8% and 30.0 ± 12.7% reduction in the total number of large neurons in the *eCN+GCP-En1/2* CKOs and *eCN-En1/2* CKOs, respectively, and no change in the *GCP-En1/2* CKOs (*Figure 6B*). Furthermore, the mediolateral distribution in both *eCN+GCP-En1/2* CKOs (*Figure 6C*) and *eCN-En1/2* CKOs (*Figure 6D*) showed a significant preferential loss in medial and intermediate CN, similar to the *Atoh1-En1/2* CKOs (*Figure 6E* and *Figure 1Q*). Also, there was an estimated 1.5 ± 0.43 fold increase in the ratio of the number of vermal PCs in the ASec and CSec to the medial eCN in *eCN-En1/2* CKOs (*Figure 6G*), although not significantly in the *eCN+GCP-En1/2* CKOs compared to controls (*Figure 6F*).

## *En1/2* promote differentiation of granule cells

Interestingly, although in the *GCP-En1/2* CKOs the number of PCs, densities of GCs and molecular layer interneurons and areas of each layer were not altered and the proportion of the area occupied by the molecular layer was conserved, as in *Atoh1-En1/2* CKOs the IGL occupied a slightly smaller proportion of the total area in the CSec and PSec of the vermis (*Figure 5—figure supplement 3A–I*). This result could mean that fewer GCs are produced in the two mutants. However, this phenotype was very mild and no changes were observed in the GC density or the GC to PC ratio in the *GCP-En1/2* CKOs compared to their control littermates (*Figure 5—figure supplement 3J–K*). We therefore used our mosaic mutant analysis approach to test whether production of GCs is altered when *En1/2* are removed from scattered GCPs at P2. Indeed, we observed a significant decrease in the

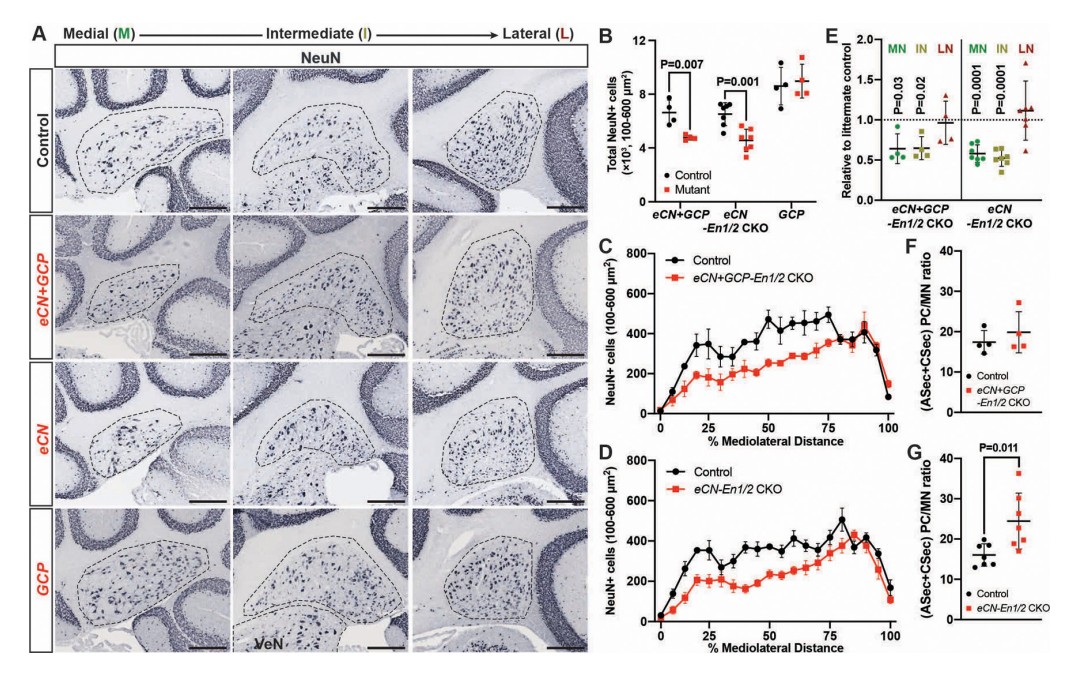

**Figure 6.** eCN numbers are reduced in *eCN+GCP-En1/2* and *eCN-En1/2* CKOs but not in *GCP-En1/2* CKOs at P30. (A) Representative immunohistochemistry for NeuN in all Dox regimens from midline (left) to lateral (right) cerebellum of P30 control and each of the cell specific *En1/2* CKOs. (B) Quantification of NeuN+ cells that are 100–600 um$^2$ (eCN size range) showing reduction in *eCN+GCP-* and *eCN-En1/2* CKOs compared to controls, but not in *GCP-En1/2* CKOs (Two-way ANOVA, $F_{(1,24)}$=10.26, p=0.0038). (C,D) Mediolateral distribution of 100–600 um$^2$ NeuN+ cells in *eCN +GCP-* and *eCN-En1/2* CKO mutants showing a loss in medial and intermediate eCN compared to controls. eCN counts are plotted as bins of 5% percent of the mediolateral distance. (*eCN+GCP-En1/2* CKO: Two-way ANOVA, $F_{(1,120)}$=63.33, p=0.0001; *eCN-En1/2* CKO: Two-way ANOVA, $F_{(1,240)}$=104.4, p=0.0001) (E) Quantification of eCN in each nucleus of mutants relative to littermate controls (medial, MN; intermediate, IN; lateral, LN) showing a significant decrease in the MN and IN in *eCN-En1/2* CKO (one-sample t-test per nucleus; MN: p=0.03, IN: p=0.02, LN: p=0.80) and *eCN +GCP-En1/2* CKO mutants (one-sample t-test per nucleus; MN: p<0.0001, IN: p<0.001, LN: p=0.437). (F,G) Quantification of the ratio of PCs (from anterior and central sectors) to eCN in the medial nucleus in *eCN+GCP-* (F) and *eCN-En1/2* CKOs (G), compared to the control littermates (Student's t-test). Significant *post hoc* comparisons are shown in the figure. Scale bar = 250 µm. VeN = vestibular nuclei.
The online version of this article includes the following figure supplement(s) for figure 6:

**Figure supplement 1.** Semi-automated eCN counting methodology.

proportion of mutant cells (GFP+) in the inner EGL and IGL compared to controls, revealing that deletion of *En1/2* leads to a mild differentiation defect in GCPs (*Figure 5—figure supplement 3L– M*).

## *En1/2* are required for survival of embryonic eCN neurons

The reduction in the number of eCN in adult *Atoh1-En1/2*, *eCN+GCP-En1/2* and *eCN-En1/2* CKOs could be due to a reduced initial production of the neurons or a normal number could be produced and then eCN are specifically lost in the medial and intermediate CN. Loss of PCs in the three mutants could similarly be due to decreased neurogenesis or subsequent loss of the neurons. To distinguish between the possibilities, we determined when a cerebellar growth defect is first detected in *Atoh1-En1/2* CKOs. Our previous analysis showed a clear growth and foliation defect in *Atoh1-En1/2* mutants at P1 (*Orvis et al., 2012*); therefore, we analyzed embryonic stages. Quantification of the cerebellar area of mid-sagittal sections at E17.5 of mutants and controls revealed a small reduction in area in *Atoh1-En1/2* CKOs compared to littermate controls (13.3 ± 5.3%) but not at E15.5 (*Figure 7A,F*). In addition, there appeared to be a delay in the formation of the earliest fissures at E17.5 (*Figure 7C,D*). We next quantified the number of eCN neurons that expressed MEIS2 in *Atoh1-TDTom-En1/2* CKOs and *Atoh1-TDTom* controls (n = 3 mice of each genotype) at E15.5 and E17.5. All MEIS2+ cells were labeled with cytoplasmic TDTom in *Atoh1-TDTom-En1/2* CKOs (*Figure 7—figure supplement 1A–B*). In control mice, 67.7 ± 3.8% of the TDTom+ cells expressed

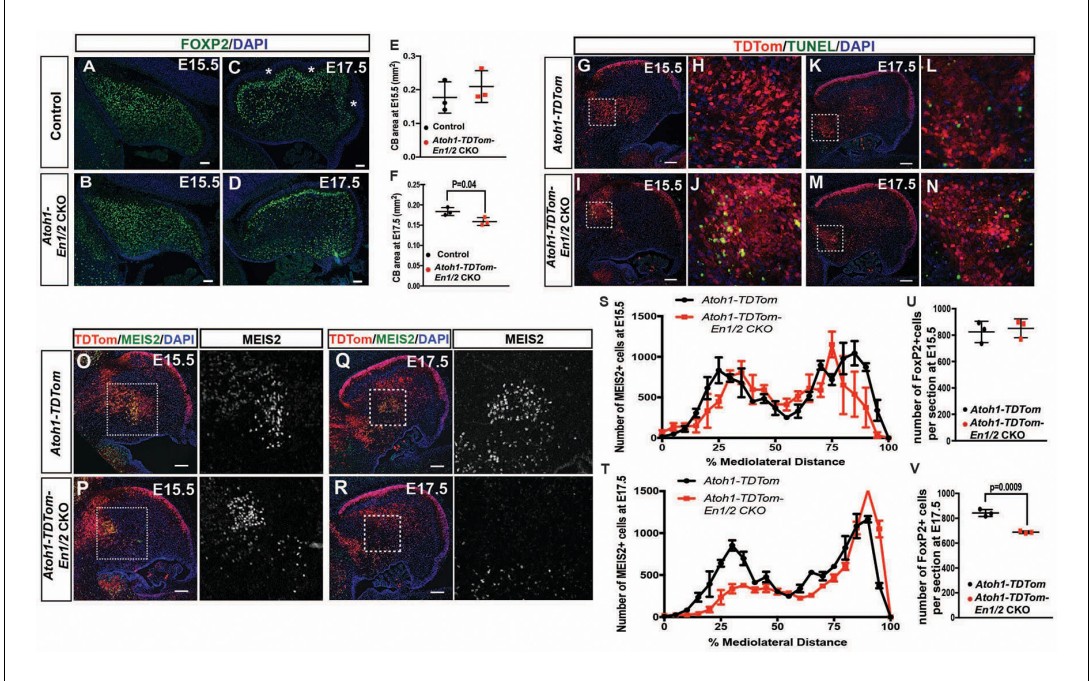

**Figure 7.** *En1/2* are required for the survival of the medial eCN and maintenance of PC numbers. (**A–F**) Immunofluorescent analysis and quantification of the cerebellar area using DAPI at E15.5 (**A,B**) and E17.5 (**C,D**) showing a significant reduction in area at E17.5 (**F**; Student's t-test) but not at E15.5 (**E**; Student's t-test) in *Atoh1-En1/2* CKOs compared to littermate controls (n = 3/condition). A slight delay in foliation was also observed at E17.5 (* base of folds). (**G–N**) Immunofluorescent analysis of TDTom and TUNEL on sagittal sections from a *Atoh1-TDTom* (**G–H, K–L**) and *Atoh1-TDTom-En1/2* CKO (**I–J, M–N**) embryo showing that apoptosis is increased in the precursors of the medial and intermediate eCN of mutants at E15.5 (**I–J**) and at E17.5 (**G–L**). (**O–R**) Representative images of sagittal sections at the level of the intermediate CN in (**S,T**) stained for MEIS2 and TDTom in *Atoh1-TDTom* (**O,P**) and *Atoh-TDTom-En1/2* CKOs (**Q,R**). (**S,T**) Quantification of the average number of MEIS2+ eCN cells in every second sagittal section along the medial-lateral axis at E15.5 and E17.5 in *Atoh1-TDTom* compared to *Atoh-TDTom-En1/2* CKOs (n = 3 for each genotype) showing reduction in the medial/intermediate eCN at E17.5. eCN counts are plotted as bins of 5% percent of the mediolateral distance (E15.5: Two-way ANOVA $F_{(1,42)}$=2.182, p=0.1471, E17.5: Two-way ANOVA, $F_{(1,42)}$=23.67, p<0.0001). (**U,V**) Quantification of PCs (FOXP2+ cells, see A-D) showing reduced PC numbers in *Atoh1-En1/2* CKOs compared to controls (n = 3 each genotype) at E17.5 (**V**) but not at E15.5 (**U**) (E15.5: Student's t-test, p=0.88). Scale bars: 100 µm.
The online version of this article includes the following figure supplement(s) for figure 7:

**Figure supplement 1.** *Atoh1-Cre* marks the rhombic lip lineage but not PCs.
**Figure supplement 2.** eCN in both *eCN+GCP-En1/2* CKOs and *eCN-En1/2* CKOs begin to die at E17.5.

MEIS2 at E15.5 and 63.5 ± 3.5% at E17.5, and a similar percentage of TDTom+ cells expressed MEIS2 in mutants (*Figure 7—figure supplement 1A–B*). None of the TDTom+ cells expressed the PC marker FOXP2, confirming the selective labeling of rhombic lip lineage at embryonic stages (*Figure 7—figure supplement 1C–D*). The TDTom+ cells in the EGL expressed PAX6, as well as a small number of cells in the forming IGL (*Figure 7—figure supplement 1E–F*). Quantification of the number of MEIS2+ cells on every other sagittal section of half of the cerebellum revealed no significant change in the total number of MEIS2+ cells at E15.5 in *Atoh1-TDTom-En1/2* CKOs compared to *Atoh1-TDTom* controls (*Figure 7O–P,S*). At E17.5, the MEIS2+ cells in the CN of mutants were significantly reduced in number by 18.5 ± 1.6% (*Figure 7Q–R,T*). The distributions of the labeled cells along the medial-lateral axis showed a specific reduction in what will likely become the intermediate and medial nuclei (*Figure 7T*).

Given the loss of eCN between E15.5 and E17.5 in *Atoh1-En1/2* CKOs, we examined whether cell death was elevated in mutants compared to controls. TUNEL assay for cell death indeed showed an increase in cell death in the region of the TDTom+ eCN population at E15.5 in *Atoh1-TDTom-En1/2* mutants compared to controls and was still observable in some regions at E17.5 (*Figure 7G–N*).

## *En1/2* loss in eCN triggers a reduction in embryonic Purkinje cell numbers

Given that eCN are lost after E15.5 in *Atoh1-En1/2* CKOs, we determined whether PCs are initially generated in normal numbers and then lost. FOXP2 labeling of PCs revealed no decrease in the number of PCs per midline section at E15.5 in *Atoh1-En1/2* CKO mutants compared to *Atoh1-TDTom* controls (*Figure 7U*). At E17.5, however, there was a significant decrease (18.3 ± 1.1%) in the number of PCs per section (*Figure 7V*). Unlike *Atoh1-TDTom-En1/2* CKOs, however, no increase in cell death was observed in either *eCN-TDTom-En1/2* CKOs or in *eCN+GCP-TDTom-En1/2* CKOs at E14.5 or E15.5 respectively (*Figure 7—figure supplement 2A,B,K,L*). At E17.5, however TUNEL+ TdTom+ cells were detected in medial regions of both the *eCN-TDTom-En1/2* CKOs and *eCN+GCP-TDTom-En1/2* CKOs (*Figure 7—figure supplement 2C,M*). Quantification of MEIS2+ eCN at E17.5, showed a significant reduction in the medial eCN in the *eCN-En1/2* CKOs compared to the littermate controls, but not in *eCN+GCP-En1/2* CKOs (*Figure 7—figure supplement 2D,N*). However, we observed a large variation in the phenotype of different mutants (both *eCN-En1/2* and *eCN+GCP-En1/2* CKOs) and that mutants which had a reduced number of eCN at E17.5 compared to their littermate controls had increased TUNEL in the medial eCN (*Figure 7—figure supplement 2F–H', P–S'*). This data shows that there is a delay in cell death in the eCN in the doxycycline-inducible *En1/2* CKOs compared to *Atoh1-En1/2* CKOs (*Figure 7I,J*), likely resulting from a delay in induction of *Cre* expression. Corresponding with the delay, we did not observe a reduction in cerebellar area or PC number per midline section in the *eCN-En1/2* or the *eCN+GCP-En1/2* CKOs at E17.5 (*Figure 7—figure supplement 2I,J,T,U*). In summary, a normal number of eCN and PCs are produced in all the *En1/2* CKOs that affect the eCN and a reduction in PC numbers is observed after eCN die.

## Genetic killing of embryonic eCN results in loss of PCs and reduced growth of the cerebellar cortex

A compelling hypothesis to explain the phenotype of the three *En1/2* CKOs that include the eCN is that when the number of eCN is reduced during embryonic development, the PCs then scale back their numbers, followed by scaling down of the cerebellar cortex growth. We tested this hypothesis by using a pharmacogenetic strategy to kill eCN. Expression of DTA was induced in the eCN using *Atoh1-tTA* and a new line of mice based on our recently described DRAGON allele (*Roselló-Díez et al., 2018*) (paper in preparation; *Figure 8A*). When *Atoh1-tTA; Atoh1-Cre; Igs7$^{TRE-LtSL-DTA/+}$* mice were administered Dox starting at E13.5 (*eCN-DTA* mice) and analyzed at P30, a significant decrease in the number of eCN was detected (41.2 ± 2.5%) compared to littermate controls (*Igs7$^{TRE-LtSL-DTA/+}$*), and unlike in *Atoh1-En1/2* CKOs, the numbers were decreased across all three nuclei (*Figure 8B,C*). Moreover, as predicted since the eCN are killed in all three CN, the area of cerebellar sections and the number of PCs in *eCN-DTA* were reduced not only in the vermis and paravermis, but also in the hemispheres at P30 (*Figure 8D,I*). Furthermore, the proportions of IGL and ML area and the ratio of PCs to GCs were preserved, showing that a scaling down of the cerebellar cortex accompanied the loss of eCN (*Figure 8K–M*). The estimated ratio of vermal ASec and CSec PCs to medial eCN, however, was increased 1.4 ± 0.3 fold in *eCN-DTA* mice (*Figure 8J*). Analysis of TUNEL in *eCN-DTA* mice at E13.5 (through E15.5) showed the expected increased cell death in TDTom+ cells in the nuclear transitory zone and region of the CN (*Figure 8N–Q*). Interestingly, PC numbers were normal at E15.5 (*Figure 8R*), demonstrating that the reduction in PCs numbers seen at P30 is not a defect in neurogenesis. Thus, similar to *En1/2* CKOs that include the eCN, when eCN are killed soon after they are generated, pre-synaptic partner PCs subsequently are reduced in number and then postnatal growth of the cerebellar cortex is proportionally decreased.

## Discussion

In summary, the results of our studies show that development of the cerebellum involves a sequence of events that includes the earliest-born neurons, the eCN, having an influence on the final number of their pre-synaptic partner PCs by supporting their survival after PC neurogenesis is complete. Previous research indicates that the PC-derived mitogenic factor, SHH, stimulates the proportional proliferation of GC, interneuron and glial progenitors and thus the level of production of all postnatally derived neurons in the cerebellar cortex (*Figure 9*) (*Corrales et al., 2006; Corrales et al., 2004;*

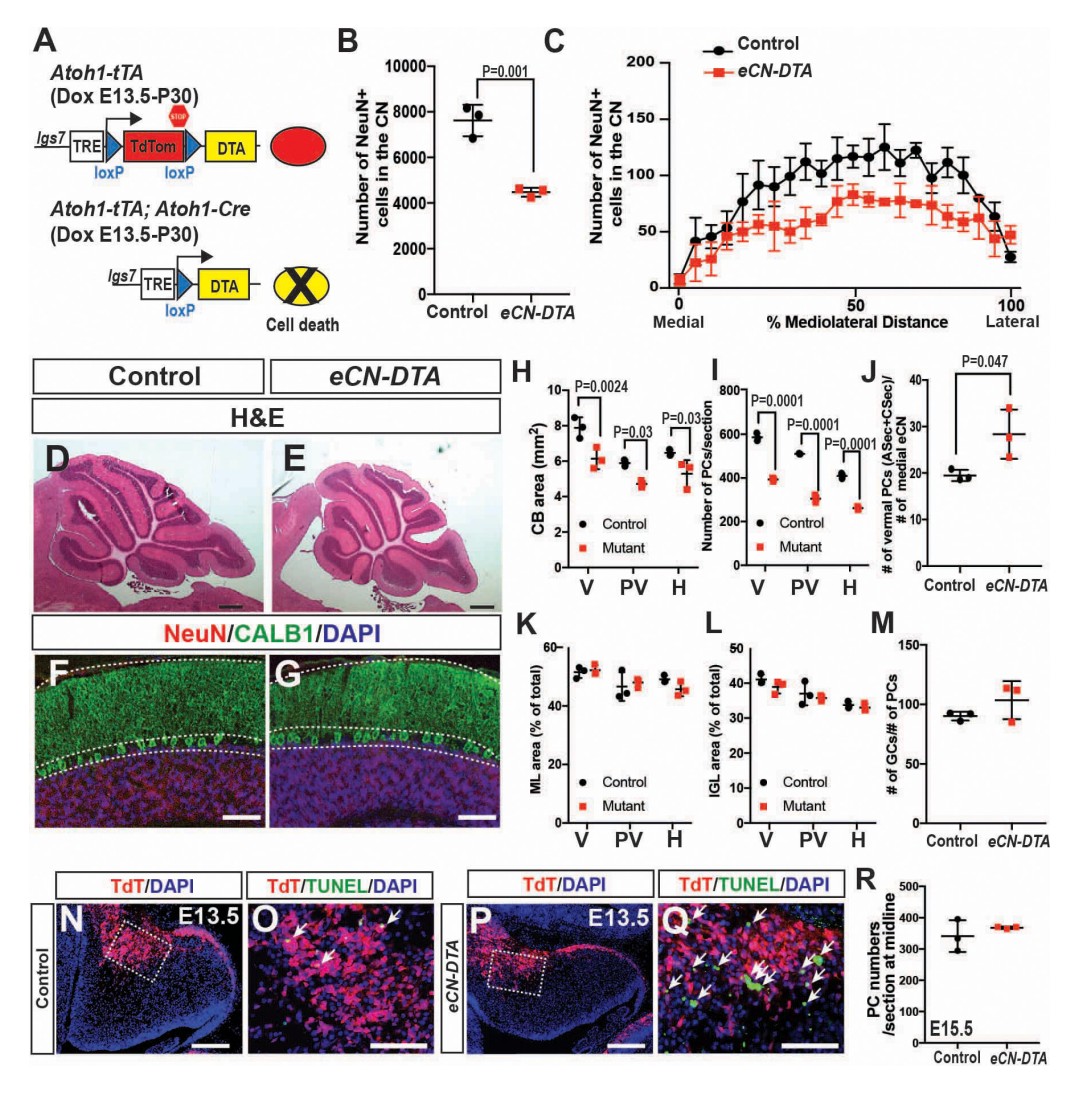

**Figure 8.** Genetic killing of embryonic eCN results in subsequent loss of PCs and reduced growth of the cerebellar cortex. (**A**) Schematic explaining the genetics used to kill PCs in all three CN. Dox-controlled and recombinase activated gene overexpression (DRAGON) allele was used in combination with the *Atoh1-tTA* and *Atoh1-Cre* to enable embryonic killing of eCN via mis-expression of DTA (Dox given from E13.5 until sacrifice). (**B**) Semi-automated quantification of the large NeuN+ cells in the eCN on every second slide showing significant reduction in eCN-DTA animals at P30 compared to their littermate controls (n = 3 animals/genotype, Student's t-test). (**C**) Cell loss was similar across the mediolateral axis (Two-way ANOVA, $F_{(1,88)}$=52.48, p=0.0001). (**D–G**) H and E analysis (**D,E**) and immunofluorescent analysis (**F,G**) of PCs (CALB1+) and GCs in the IGL (NeuN+) on midsagittal sections showing reduction in cerebellar area but conserved cytoarchitecture. (**H–I**) Quantification of cerebellum area (**H**) and PC numbers (**I**) showing a significant reduction at all mediolateral levels at P30 in *eCN-DTA* animals compared to control littermates (cerebellum area: Two-way ANOVA, $F_{(1,4)}$=19.47, p=0.0116, PC numbers: Two-way ANOVA, $F_{(1,4)}$=38.86, p=0.0034) (V: vermis, PV: paravermis, H: hemispheres). (**J**) Estimated ratio of vermal ASec+CSec PCs to the medial eCN showing it is increased in the *eCN-DTA* animals compared to control littermates (n = 3/genotypes, Student's t-test). (**K–M**) Quantification showing relative ratios of ML (**K**) and IGL (**L**) area compared to total sector area and the ratio of the number of GCs to PC (**M**, data quantified in the vermis) are unchanged. (**N–Q**) TUNEL staining for cell death showing an increase in the nuclear transitory zone at E13.5 in *eCN-DTA* animals compared to controls (n = 3). (**R**) Quantification of PC numbers (FOXP2+) at E15.5 on midsagittal sections showing that PCs are produced at a normal number in *eCN-DTA* animals (n = 3). Significant *post hoc* comparisons are shown in the figure. DTA: diphtheria toxin fragment A. Scale bars: D-E: 500 µm, F-H and N-Q 100 µm.

De Luca et al., 2015; Fleming et al., 2013; Lewis et al., 2004; Parmigiani et al., 2015; Wojcinski et al., 2017). This progression of neurogenesis and cell survival that maintains the overlying circuit constitutes a means by which developmental defects affecting the eCN will lead to adaptations that allow a near normal cytoarchitecture to form in the cerebellar cortex. We hypothesize that once the PCs are born their numbers are then pruned or scaled back to be proportional to the

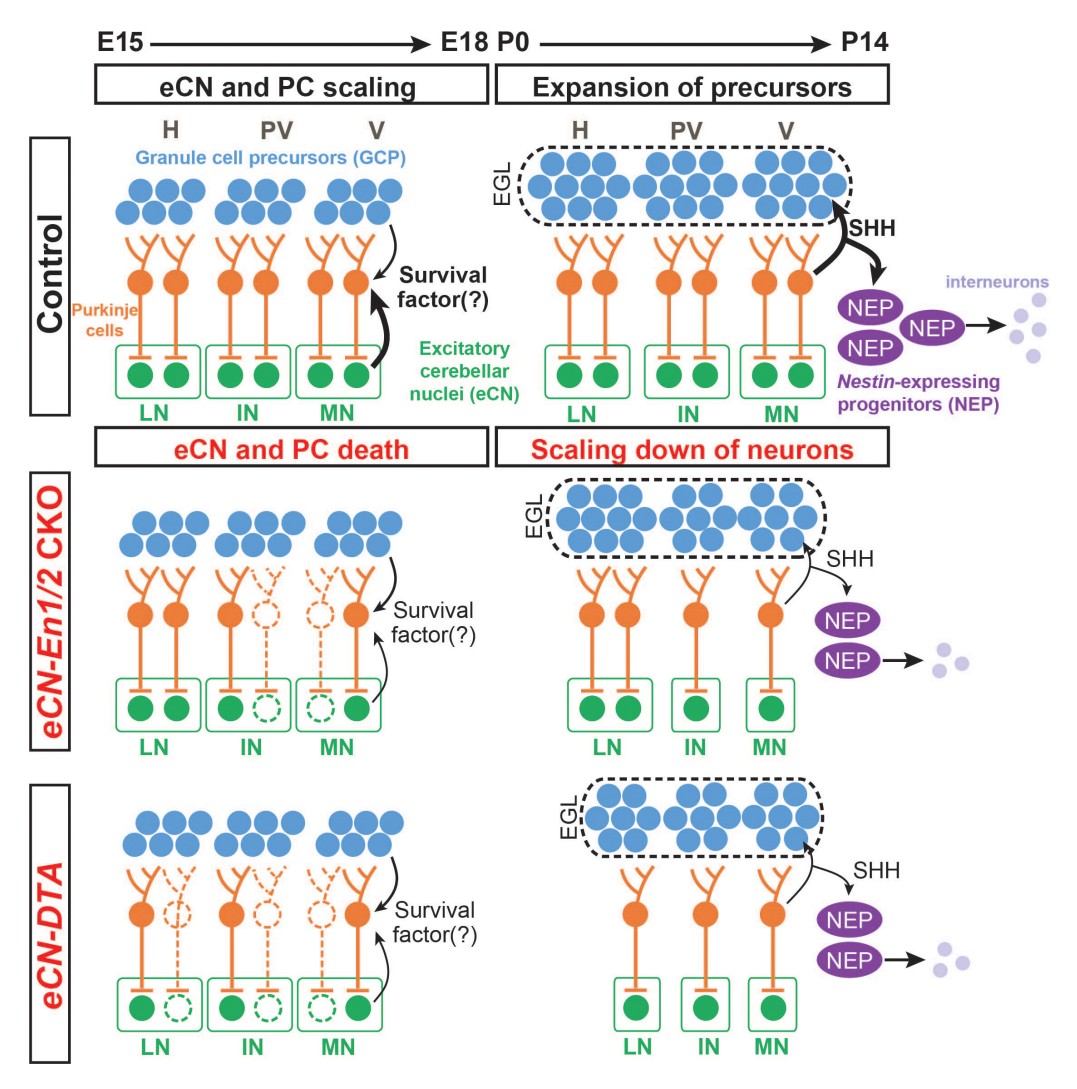

**Figure 9.** Schematic representation of the proposed mechanisms for scaling of neuron numbers in the cerebellum. In normal conditions, the first born cerebellar neurons, the eCN, determine the number of their presynaptic partner PCs based on expression of a survival factor. PCs then in turn regulate the number of GCPs and nestin-expressing progenitors (NEPs) that generate the GCs and interneurons, respectively through expression of SHH. In *eCN-En1/2* CKOs involving the eCN, eCN specifically in the medial and intermediate CN die after E15.5 and then their presynaptic partner PCs are lost, resulting in a smaller vermis and paravermis while preserving scaling of neuron numbers in the cortex. In *eCN-DTA* animals, eCN in all nuclei are killed and therefore growth of all regions of the cerebellum is reduced but cortex neurons are scaled. H: hemisphere, PV: paravermis, V: vermis, LN: lateral nucleus, IN: intermediate nucleus, MN: medial nucleus.

The online version of this article includes the following figure supplement(s) for figure 9:

**Figure supplement 1.** RNA in situ hybridization of *Bdnf* shows expression in a subpopulation of eCN at E15.5.

number of eCN and this involves the neurotrophin theory (*Levi-Montalcini and Hamburger, 1951*; *Snider, 1994*) or secretion of a survival factor by eCN.

The results of our study show that if the number of PCs is reduced around birth (either in *En1/2* CKOs or due to killing of PCs using DT), then the other neurons generated postnatally in the cerebellar cortex are scaled back in numbers to be proportional to the number of PCs, and thus producing a normally proportioned cytoarchitecture (*Figure 9*). Considering the cerebellar cortex as a multilayer structure comprised of unitary tiles containing a single PC which is able to support and specify postnatally-derived cells that contribute to its local circuit, the size of the PC pool can be thought of as a predictor of total postnatal cerebellum growth capacity. As the PCs redistribute into a monolayer across the cerebellar cortex after birth, the smaller population of PCs in *En1/2* CKOs

and *eCN-DTA* mice is not able to spread out to form the normal length of the PC layer, since they have the same density at P30 as controls. As SHH expressed by PCs is required for expansion of the two proliferating precursor populations in the postnatal cerebellum (GCPs and nestin-expressing progenitors)(*Corrales et al., 2006*; *De Luca et al., 2015*; *Fleming et al., 2013*; *Lewis et al., 2004*; *Wojcinski et al., 2017*), the scaling back in *En1/2* CKOs and *eCN-DTA* mutants of granule cells and interneurons is likely due to reduced progenitor proliferation because the amount of SHH present in the mutant cerebella is reduced. In this context the cerebellar cortex could be considered as made up of repeated developmental tiles of neurons, and it is interesting to note that our results show that PCs regulate the numbers of their presynaptic neurons that are generated by neurogenesis.

Based on our finding that PCs scale down their numbers soon after eCN die in *En1/2* CKOs and *eCN-DTA* mice, our study has provided evidence that the survival of PCs is dependent on eCN, suggesting a retrograde regulation of cell viability (*Cowan et al., 1984*). The timing of innervation of eCN by PCs fits with this proposal as axons from PCs extend into the CN by E15.5 (*Sillitoe et al., 2009*). Furthermore, in vitro assays have shown that PC survival is dependent on cell-cell interactions (*Baptista et al., 1994*) or on neurotrophins (*Lärkfors et al., 1996*; *Mount et al., 1995*), and mouse mutants have shown that *Bdnf* and $p75^{NTR}$ null mutants exhibit cerebellar growth and foliation defects with some similarities to *Atoh1-En1/2* mutants (*Carter et al., 2003*; *Carter et al., 2002*; *Schwartz et al., 1997*). Based on gene expression in the Allen Institute Database and our analysis in *Atoh1-TDTom* mice, *Bdnf* is expressed specifically in the CN at E15.5 (*Figure 9—figure supplement 1*); thus, it is a good candidate for one of the eCN-derived survival factor for PCs. *Gdnf* is also detected in the CN after birth, suggesting that it could also contribute to long-term survival of PCs in the cerebellum.

It appears that scaling of numbers between the eCN and PCs is not as precise as the scaling within the cerebellar cortex in all the mutants with a decreased number of embryonic eCN (*En1/2* CKOs and *eCN-DTA* mutants). It is possible that the imprecise proportions of eCN and PCs accounts for the motor behavior deficits we observed in *Atoh1-En1/2* CKOs, although the smaller size of the cerebellum (~30% reduction) could also contribute as it might not be able to support all the long range neural circuits needed for full motor function. In a previous study an ~20% reduction in the cerebellum without disruption of the cytoarchitecture did not lead to obvious motor defects, however the eCN and other cell types were not quantified and the reduction in size was less (*Wojcinski et al., 2017*). The higher than expected ratio of PCs in the vermis to the medial eCN might reflect the way in which scaling is established. GC and interneuron numbers depend on a precise and deterministic proliferative response to SHH downstream of PC number (*Corrales et al., 2006*; *De Luca et al., 2015*; *Fleming et al., 2013*; *Lewis et al., 2004*; *Wojcinski et al., 2017*). In contrast, we propose that PC number depends on stochastic retrograde survival factors from the eCN. Moreover, PC survival could be complicated by the convergence ratio of ~40 PCs to each eCN (*Person and Raman, 2011*).

As our model asserts that eCN neurons maintain the survival of their presynaptic PC neurons (*Figure 9*), we predict that eCN in the medial and intermediate nuclei that are lost in the greatest numbers in the *Atoh1-En1/2* CKOs are preferentially innervated by PCs in the vermis and paravermis lobules, respectively, that exhibit the most diminution. Consistent with this, in *eCN-DTA* mice where eCN are killed evenly throughout the cerebellum, all regions of the cerebellum are similarly reduced in size. Our finding that the foliation defects in *Atoh1-En1/2* CKOs are distinct from *eCN-* and *eCN+GCP-En1/2* CKOs could then be explained by the different *Cre* transgenes used targeting slightly different subsets of eCN or because of the delay in Cre expression using the tTA system. Determination of the projection pattern of PC axons into the CN showed that PCs of the vermis central sector, where lobule growth and PC depletion are most pronounced in *Atoh1-En1/2* CKOs, do indeed preferentially innervate the posterior region of the medial CN that was also preferentially reduced in *Atoh1-En1/2* CKOs. In a complementary fashion, the PCs of the ASec preferentially project to the anterior medial nucleus and part of the intermediate nucleus. Detailed mapping of the eCN that die in each *En1/2* CKO and the innervation patterns of PCs to spatial regions of each CN will be required to fully understand the phenotypes.

The first detected defect in *Atoh1-En1/2* CKOs is death of eCN starting at ~E15.5, indicating that *En1/2* are required in eCN for their survival in the embryo as EN1/2 protein is fully lost in the mutants by E15.5. This survival function of *En1/2* is similar to the proposed function of the genes in other neurons in the brain (*Fox and Deneris, 2012*; *Sonnier et al., 2007*). Conditional knockout of

*En1/2* in embryonic postmitotic serotonergic (5-HT) neuron precursors of the dorsal Raphe nucleus (DRN) results in a normal number of 5-HT neurons being initially generated, but the cells then become disorganized through abnormal migration and then a subset of cells undergo significant cell death around birth (*Fox and Deneris, 2012*). As *En1/2* are ablated in GCPs as well as eCN in *Atoh1-En1/2* mutants, we expected that some of the postnatal growth defect would be due to a requirement for *En1/2* after birth in the GCPs. However, we found that eCN- and not *GCP-En1/2* CKOs have a growth defect and loss of eCN. Mosaic analysis of the role of *En1/2* in GCPs revealed only a mild function in promoting GCP differentiation, consistent with the expression of the EN proteins after birth mainly in the inner EGL where granule cells begin differentiating. This role for *En1/2* is consist with our recent finding that deletion of *En1/2* in GCPs along with over-activation of SHH signaling results in a small increase in the proportion of mutant cells that remain in the proliferative GCP pool (*Tan et al., 2018*).

In addition to the RL-lineage, *Atoh1-Cre* deletes *En1/2* in the mossy fiber precerebellar nuclei, some of which express *En1/2*. Most of the mossy fiber precerebellar nuclei, but not the inferior olive, are also targeted in *eCN-En1/2* CKOs and *eCN-DTA* mice. Thus, it is possible that some of the phenotypes (behavioral and cellular) seen in both mutant types are a secondary (cell-nonautonomous) effect of loss of *En1/2* or killing with DT of neurons outside the cerebellum. If this is the case, one interpretation consistent with our hypothesis would be that eCN survival is dependent on neurotrophins secreted by precerebellar nuclei neurons. If *En1/2* are required in a subset of mossy fiber neurons for their survival, then their loss in *En1/2* CKOs or *eCN-DTA* mice could result in death of their partner eCN. The scaling back of PCs, however, would not likely be due to loss of mossy fiber neurons since PCs do not project to them.

In conclusion, *En1* and *En2* are required in the eCN after they are produced by the RL for the survival of a subset of medial/intermediate CN neurons, possibly the ones that normally express EN1/2 after E15.5. Loss of a subset of medial/intermediate eCN neurons results in cell non-autonomous loss of the PCs that are their presynaptic partners in the vermis and paravermis (*Figure 9*). When eCN are killed in all three nuclei in *eCN-DTA* mutants, PC numbers are reduced in all regions of the cerebellum (*Figure 9*). The size of the cerebellar cortex is then scaled back to be proportional to the number of PCs that remain in each region, producing a normal appearing cytoarchitecture. The PCs do not appear to scale back fully to the number of the remaining eCN. Our results have important implications for how scaling of neurons in all brain regions could be regulated by a combination of mitogens that stimulate proliferation of progenitors (neurogenesis) and survival factors that promote the survival of the neurons that are produced (neurotrophic theory) to help match synaptic partner neuron proportions. These proposed mechanisms of scaling could act in local or long distance brain circuits, as well as during evolution of mammals, possibly accounting for the parallel large expansion of the lateral eCN (dentate) and hemispheres in primates.

## Materials and methods

### Key resources table

| Reagent type (species) or resource | Designation | Source or reference | Identifiers | Additional information |
|---|---|---|---|---|
| Genetic reagent (*M. musculus*) | *En1$^{lox}$* | *Sgaier et al., 2007* | Jackson Labs stock no: 007918 RRID:MGI:3789091 | |
| Genetic reagent (*M. musculus*) | *En2$^{lox}$* | *Cheng et al., 2010* | Jackson Labs stock no: 007925 RRID:MGI:3831084 | |
| Genetic reagent (*M. musculus*) | Atoh1-Cre | *Matei et al., 2005* | Jackson Labs stock no: 011104 RRID:MGI:3775845 | |
| Genetic reagent (*M. musculus*) | R26$^{LSL-TDTom}$ (ai14) | *Madisen et al., 2010* | Jackson Labs stock no: 007909 RRID:MGI:3809524 | |

*Continued on next page*

*Continued*

| Reagent type (species) or resource | Designation | Source or reference | Identifiers | Additional information |
|---|---|---|---|---|
| Genetic reagent (*M. musculus*) | *R26^{LSL-nTDTom}* (ai75D) | | Jackson Labs stock no. 025106 RRID:MGI:5603432 | |
| Genetic reagent (*M. musculus*) | *R26^{LSL-lacZ}* | | Jackson Labs stock no. 002073 RRID:MGI:1861932 | |
| Genetic reagent (*M. musculus*) | *Pcp2^{Cre}* | *Zhang et al., 2004* | | Made by H. Saito and N. Suzuki, provided by P. Faust. |
| Genetic reagent (*M. musculus*) | *TetO-Cre* | *Perl et al., 2002* | Jackson Labs stock no. 006234 RRID:MGI:3770672 | |
| Genetic reagent (*M. musculus*) | *Atoh1-tTA* | described here | | See Materials and methods |
| Genetic reagent (*M. musculus*) | *En2-HA* | described here | | See Materials and methods |
| Genetic reagent (*M. musculus*) | *Igs7^{TRE-LtSL-DTA/+}* | To be described elsewhere, modified from *Roselló-Díez et al. (2018)* | | |
| Antibody | Rabbit anti-MEIS2 (K846) | *Mercader et al., 2005* | Gift from Dr. Miguel Torres (CNIC, Spain) | 1:3000 for embryonic tissue, 1:2000 for adult tissue |
| Antibody | Goat anti-FOXP2 | Everest Biotech | EB05226 RRID:AB_2107112 | 1:1000 |
| Antibody | Rabbit anti-panEN | *Davis et al., 1991* | | 1:500 |
| Antibody | Rabbit anti-Calbindin | Swant Inc | CB38 RRID:AB_2721225 | 1:1000 |
| Antibody | Mouse anti-Calbindin | Swant Inc | 300 RRID:AB_10000347 | 1:500 |
| Antibody | Mouse anti-NeuN | Millipore | MAB377 RRID:AB_2298772 | 1:1000 |
| Antibody | mouse anti-Parvalbumin | Millipore | MAB1572 RRID:AB_2174013 | 1:500 |
| Antibody | rat anti-HA | Sigma | 11867423001 | 1:1000 |
| Recombinant DNA reagent | AAV5-EF1a-DIO-mCherry | UNC Chapel Hill Vector Core | | |
| Commercial assay/Kit | In situ Cell Death Detection Kit | Roche (via Sigma) | 11684795910 | |
| Chemical compound/Drug | Doxycycline hyclate | Sigma | D9891 | 0.02 mg/mL in drinking water |

## Mouse strains

All animal experiments were performed in accordance with the protocols approved and guidelines provided by the Memorial Sloan Kettering Cancer Center's Institutional Animal Care and Committee (IACUC). Animals were given access to food and water ad libitum and were housed on a 12 hr light/dark cycle.

All mouse lines were maintained on a Swiss-Webster background: *En1^{lox}* (*Sgaier et al., 2007*), *En2^{lox}* (*Cheng et al., 2010*), *Atoh1-Cre* (*Matei et al., 2005*), *R26^{LSL-TDTom}* (ai14, Jackson labs stock no: 007909) (*Madisen et al., 2010*), *R26^{LSL-nTDTom}* (ai75D, Jackson labs stock no: 025106), *R26^{LSL-lacZ}* (Stock no. 002073, The Jackson Laboratories), *Pcp2^{Cre}* (*Zhang et al., 2004*), *TetO-Cre* (Stock no. 006234, The Jackson Laboratories) (*Perl et al., 2002*).

*Atoh1-tTA* mouse line was generated by replacing the CreERT2 cassette of the *Atoh1-CreERT2* (*Machold and Fishell, 2005*) with the coding sequence of tTA2s (including the polyA) from the pTet-Off Advanced vector (Clontech). *En2-HA* mouse line was generated by inserting a six amino acid

(GlyLysSerAspSerGlu) linker and three consecutive HA tags (*Fu et al., 2009*) to the C-terminus of the *En2* gene, replacing the stop codon, via CRISPR-Cas9 using the gRNA 5'- GGCAAGTCGGACAGC-GAGTA −3'. One mouse line was confirmed with sequencing to have the correct modification. Homozygous (*En2$^{HA/HA}$*) and *En2$^{HA/-}$* animals did not have any obvious cerebellar phenotype. The *Igs7$^{TRE-LtSL-DTA/+}$* mouse line was generated using our recently described DRAGON allele (*Rose-lló-Díez et al., 2018*) and will be described elsewhere (paper under preparation, *Figure 8A*).

Noon of the day that a vaginal plug was discovered was designated developmental stage E0.5. Both sexes were used for all experiments and no randomization was used. Exclusion criteria for data points were sickness or death of animals during the testing period. For behavioral testing, investigators were blinded to genotype during the data collection and analyses.

## Doxycycline treatment

Doxycycline (Dox) hyclate (Sigma D9891) was added to the drinking water at a final concentration of 0.02 mg/mL and replaced with fresh Dox every 3–4 days.

## Behavioral testing

One-month-old (Control: n = 8 and *Atoh1-En1/2* CKOs: n = 9) or 3-month-old (Control: n = 16 and *Atoh1-En1/2* CKOs: n = 11) were used to for all of the motor behavior testing described below.

### Rotarod

Analysis was performed three times a day on three consecutive days. Animals were put on an accelerating rotarod (47650, Ugo Basile), and were allowed to run till the speed reached 5 rpm. Then, the rod was accelerated from 10 to 40 rpm over the course of 300 s. Latency to fall was recorded as the time of falling for each animal. Animals rested for 10 min in their home cage between each trial.

### Grip Strength

A force gauge with a horizontal grip bar (1027SM Grip Strength meter with single sensor, Columbus Instruments) was used to measure grip strength. Animals were allowed to hold the grip bar while being gently pulled away by the base of their tail. Five measurements with 5 min resting intervals were performed for each animal and the data was reported as the average of all trials, normalized to the mouse's weight (Force/gram).

### Footprinting Analysis

After painting the forefeet and hindfeet with red and blue nontoxic acrylic paint (Crayola), respectively, animals were allowed to walk on a strip of paper. Experiments were performed in a 50 cm long and 10 cm wide custom-made Plexiglass tunnel with a dark box at one end. Each mouse was run through the tunnel three times and the distance between the markings were measured and averaged.

## Genotyping

The DNA primers used for genotyping each allele and transgene were as follows: *En1$^{lox}$* (*Orvis et al., 2012*), *En2$^{lox}$* (*Orvis et al., 2012*), *Cre* (*Atoh1-Cre* and *Pcp2$^{Cre}$*): 5'-TAAAGATATC TCACGTACTGACGGTG, 5'-TCTCTGACCAGAGTCATCCTTAGC; *R26$^{TDTom}$* (nuclear and cytoplasmic): 5'-CTGTTCCTGTACGGCATGG, 5'-GGCATTAAAGCAGCGTATCC; *tTA* (*Atoh1-tTA*): 5'-G TCAATTCCAAGGGCATCGGTAAAC, 5'-AATTACGGGTCTACCATCGAGGGC; *tetO-Cre*: 5'-AGC TCGTTTAGTGAACCGTCAG, 5'-CTGTCACTTGGTCGTGGCAGC.

## Tissue preparation and histology

The brains of all embryonic stages were dissected in cold PBS and immersion fixed in 4% PFA for 24 hr at 4°C. All postnatal stages were transcardially perfused with PBS followed by 4% PFA, and brains were post-fixed in 4% PFA overnight, and washed three times in PBS. Specimens for paraffin embedding were put through an ethanol-xylene-paraffin series, then embedded in paraffin blocks, and sectioned to 10 μm on a microtome (Leica Instruments). Specimens for cryosectioning were placed in 30% sucrose/PBS until they sank, embedded in OCT (Tissue-Tek), frozen in dry ice cooled isopentane, and sectioned at 14 μm on a cryostat unless otherwise stated (Leica, CM3050S).

## Immunohistochemistry

Sections of cryosectioned tissues were immersed in PBS for 10 min. After these initial stages of processing, specimens were blocked with blocking buffer (5% Bovine Serum Albumin (BSA, Sigma) in PBS with 0.2% Triton X-100). Primary antibodies in blocking buffer were placed on slides overnight at 4°C, washed in PBS with 0.2% Triton X-100 (PBST) and applied with secondary antibodies (1:1000 Alexa Fluor-conjugated secondary antibodies in blocking buffer) for 1 hr at room temperature. Counterstaining was performed using Hoechst 33258 (Invitrogen). The slides were then washed in PBST, then mounted with a coverslip and Fluorogel mounting medium (Electron Microscopy Sciences).

Hematoxylin and Eosin (H and E) staining was performed for histological analysis and cerebellar area measurements, except at E17.5 where DAPI staining was used. Nissl staining using cresyl violet was used for stereology analysis in the eCN in adult brains.

For eCN counting in *eCN+GCP-En1/2* CKO, *eCN-En1/2* CKO, and *GCP-En1/2* CKO P30 animals, we used nickel enhanced DAB (Ni-DAB) immunohistochemistry for NeuN. All steps were performed at room temperature, unless specified. Cryo-sectioned tissue were washed in PBS for 5 min for three times (hereafter PBS washes). After washes, antigen retrieval was achieved by immersing sections in sodium citrate buffer (10 mM sodium citrate with 0.05% Tween-20, pH 6.0) at 95°C for 40 min. After antigen retrieval, sections were washed in PBS and treated with 50% methanol with 0.03% $H_2O_2$ for 1 hr. Sections were then washed with PBS and applied with anti-NeuN primary antibody solution in blocking buffer (1:1000 dilution in 0.4% PBST with 5% BSA) overnight at 4°C. After washing in PBS the following day, biotinylated anti-mouse secondary antibody (Vector Labs, Burlingame, CA, USA; BA-9200) in blocking buffer (1:500) were applied for 1 hr, followed by PBS washes. Vectastain Elite ABC HRP solution (Vector Labs, Burlingame, CA, USA; PK-6100) in blocking buffer (1:500 for A and B) were applied for 1 hr, followed by PBS washes. Sections were washed in 0.175 M sodium acetate (in $dH_2O$) and incubated in Ni-DAB solution (0.02% 3,3'-Diaminobenzidine tetrahydrochloride (Sigma-Aldrich, D5905), 2.5% $H_2O_2$, and 2.5% nickel sulfate in 0.175 M sodium acetate) for 20 min. Ni-DAB reaction was stopped by washing sections in 0.175 sodium acetated followed by PBS washes. After dehydration and defatting ($dH_2O$-70% ethanol-95%–95 %-100%-100%-xylene-xylene-xylene, one minute each) slides were mounted with a coverslip and DPX mounting medium (Electron Microscopy Sciences).

## Antibodies

The following antibodies were used at the listed concentrations: Rabbit anti-MEIS2 (1:3000 for embryonic tissue, 1:2000 for adult sections, K846 rabbit polyclonal; *Mercader et al., 2005*), Goat anti-FOXP2 (Everest Cat no: EB05226, 1:1000), Rabbit anti-panEN; 1:200 (*Davis et al., 1991*), anti-Calbindin (Swant Inc Cat No: CB38, 1:1000 rabbit, Cat No: 300, 1:500 mouse), Mouse α-NeuN (Millipore Cat no: MAB377, 1:1000), mouse α-Parvalbumin (Millipore Cat no: MAB1572, 1:500), rat α-HA (Sigma Cat no: 11867423001).

## RNA in situ hybridization

Probes were in vitro transcribed from PCR-amplified templates prepared from cDNA synthesized from postnatal whole brain or cerebellum lysate. The primers used for PCR amplification were: *Slc17a6*: Forward: 5'-AGACCAAATCTTACGGTGCTACCTC-3', Reverse: 5'-AAGAGTAGCCATC TTTCCTGTTCCACT-3' *Slc6a5*: Forward: 5'- GTATCCCACGAGATGGATTGTT-3', Reverse: 5'- CCA TACAGAACACCCTCACTCA-3' *Bdnf*: Forward: 5'-CGACGACATCACTGGCTG-3', Reverse: 5'-CGGCAACAAACCACAACA-3'. Primers were flanked in the 5' with SP6 (antisense) and T7 (sense) promoters. Specimen treatment and hybridization were performed as described previously (*Blaess et al., 2011*).

## Image acquisition and analysis

All images were collected with a DM6000 Leica fluorescent microscope or a Zeiss Axiovert 200 with Apotome or NanoZoomer Digital Pathology (Hamamatsu Photonics) and processed using ImageJ (NIH) or Photoshop (Adobe). Image quantification was performed with ImageJ.

For the cerebellar, sector, IGL, and ML area, H and E stained slides were used. A region of interest was defined by outlining the perimeter of the outer edges of the region quantified. Threesagittal sections/brain at each mediolateral level were analyzed, and values were averaged for each brain.

Cell counts were obtained using the Cell Counter plugin for ImageJ (NIH). The average number of total Calbindin positive PCs per section was determined from three sections per animal in each region (vermis or midline, paravermis and hemispheres). PC density was calculated by dividing the average number of PCs by the average length of the PCL in each region (three sections per animal per region). Granule cell density was measured by counting the NeuN+ cells in a 40x frame of the IGL from three sections/animal and region and dividing the number of cells by the area counted. GC density for different vermis sectors was obtained from lobule 3 (anterior), lobules 6–7 (central), and lobule 9 (posterior) from three vermis sections per animal. GC numbers per section in the vermis were calculated by multiplying the average GC density by the average area of the IGL (n = 3 sections per animal). Interneuron densities were calculated as the average number of Parvalbumin+ cells in the ML of one section for lobule 3 (anterior), 6–7 (central) or 9 (posterior) divided by the average ML area per section (n = 5 sections per animal). The total number of ML Parvalbumin+ interneurons was extrapolated by multiplying the average density of the three lobules times the average total ML area.

Ratios of the the number of GCs to PCs, Parvalbumin+ cells in the ML to PCs and GCs to Paravalbumin+ cells in the ML were calculated using the total numbers calculated above per average vermis sections. The ratio of PCs (vermal ASec+CSec) to eCN (medial) was obtained by dividing the extrapolated number of PCs in the ASec and the CSec of a half vermis (PC average number in the ASec and CSec on one midline section multiplied by half the number of 14 μm-thick sections in a half vermis) divided by the number of medial eCN present on every second section of half a cerebellum using our two different methods (semi-automated eCN counting of cell type specific *En1/2* CKOs and *eCN-DTA* mutants or stereology for *Atoh1-En1/2* cKOs).

## Stereology

Brains were paraffin embedded, as described above, and sectioned coronally at 10 μm. The sections were Nissl stained and every other section was analyzed to prevent double counting neurons split in serial sections. For each analyzed section in sequential order from rostral to caudal, the section perimeter of the CN on one half of the cerebellum was traced and each large projection neuron was registered. Neurons were categorized by nucleus at each coronal level by reference to coronal sections at that level of the cerebellum in an adult mouse brain atlas (*Paxinos and Franklin, 2001*). Sections were then aligned into a 3D representation in NeuroLucida.

## Semi-automated eCN counting methodology

For eCN counting using NeuN labeled slides, every other section was analyzed using a semi-automated quantification. Slides were scanned with a slide scanner (NanoZoomer Digital Pathology) at 20X, and images containing the eCN from each section was exported at 10X resolution. eCN from each image was cropped using Photoshop and saved as TIFF format. Using a batch 'analyze particle' process in ImageJ, each image was converted to 8-bit, sharpened, despeckled, and outliers (radius = 5, threshold = 30) were removed. Particles sized 100–600 $um^2$ from each preprocessed image was quantified (*Figure 6—figure supplement 1*). The size range was determined by analyzing the area of NEUN and nTDTom double positive cells or NEUN positive/nTDTom negative cells in an *Atoh1-Cre/+; R26^{LSL-nTDTom/+}* animal (*Figure 6—figure supplement 1*), which marks eCN neurons. To validate the reliability of the automated counts, manual counting of NeuN positive cells and semi-automated counting were compared (*Figure 6—figure supplement 1*). To determine the medial, intermediate and lateral CN neuron number, we compared the sagittal sections stained with NeuN to an adult mouse brain atlas (*Paxinos and Franklin, 2001*). Sagittal sections matching lateral 0.48 mm to 0.72 mm in the atlas were counted as medial CN, lateral 0.84 mm to 1.80 mm as intermediate (includes the dorsolateral medial CN, but not lateral CN), and lateral 1.92 mm to 2.64 mm as lateral CN. On average, the first 20% of the sections from midline were medial CN, 20–70% were intermediate CN and 70% from midline to most lateral section were lateral CN.

## Stereotactic injection

One-month-old mice were anesthetized by isoflurane inhalation and head fixed in a stereotactic frame (David Kopf Instruments) with an isoflurane inhaler. The head of the animal was shaved, disinfected with ethanol and betadine, and a midline incision was made from between the eyes to the back of the skull. Coordinate space for the system was calibrated by recording the coordinates of Bregma and Lambda. After a small hole was drilled in the skull over the injection site, the needle was robotically injected into the cerebellum position specified by atlas coordinate (Neurostar Stereo-Drive). For targeting the ASec and PSec, coordinates of the injection were −6.00 mm from Bregma, −2.2 mm deep from dura, and 8.5 mm from Bregma, 4.2 mm deep from dura, respectively. One µL of $10^{12}$Tu/mL AAV5 virion in PBS was injected in 20 millisecond pulses at 10 psi with a Picospritzer. The needle was left in situ for 5 min, and then removed by 50 µm increments over about a minute. The scalp was then sealed by Vetbond adhesive, 0.1 µg/g of buprenorphine was administered for postoperative analgesia, and the animal was placed in a heated chamber for recovery. Brains from these animals were analyzed 1.5 weeks after surgery. Brains were processed for cryosectioning as above but were sectioned at 30 µm. To establish the accuracy of the injections to the appropriate sectors, $R26^{TDTom/TDTom}$ P30 animals were injected stereotactically with either AAV5-pgk-Cre virus or trypan blue dye in the midline of either the ASec (lobules 3–5) or CSec (lobules 6–7). Whole mount imaging of TDTom signal verified that the transduced cells remained restricted to the injection area and spread minimally in the lateral axis beyond the vermis. The tracing was done by injecting AAV5-EF1a-DIO-mCherry virus into $Pcp2^{Cre/+}$ animals. The AAV-EF1a-DIO-mCherry plasmid was constructed by Karl Deisseroth and virus was packaged as serotype five and prepared to $10^{12}$ Tu/mL by the UNC Chapel Hill Vector Core.

## Tissue delipidation for tracing analysis

After perfusion, postfix, and PBS washes as described above, brains were delipidated with a modified Adipo-Clear procedure (*Chi et al., 2018*) to enhance immunolabeling in heavily myelinated regions. The brain samples were washed with a methanol gradient (20%, 40%, 60%, 80%) made by diluting methanol with B1n buffer (0.1% Triton X-100/0.3 M glycine in H2O, pH 7.0), 30 min each step; then washed in 100% methanol for 30 min twice, and reverse methanol/B1n gradient (80%, 60%, 40%, 20%) for 30 min each; then in B1n buffer for 30 min twice. After delipidation, samples were washed in PBS and sectioned with a vibratome at 50 µm. The free-floating sections were immunolabeled for mCherry for analysis of PC projections.

## Statistical analysis

Prism (GraphPad) was used for all statistical analyses. Statistical comparisons used in this study were Student's two-tailed t-test and Two-way analysis of variance (ANOVA), followed by post hoc analysis with Tukey's test for multiple comparisons. Relevant F-statistics and the significant p-values are stated in the figure legends and the p-values of the relevant *post hoc* multiple comparisons are shown in the figures. Results of the non-significant comparisons and a summary of all of the statistical analyses performed can be found in *Figure 1—source data 1*. The statistical significance cutoff was set at p<0.05. Population statistics were represented as mean ± standard deviation (SD) of the mean, except for the behavioral data presented in *Figure 2*, *Figure 2—figure supplement 1* and the mediolateral eCN number distribution graphs shown in *Figures 6* and *8*. Population statistics in these figures are represented as mean ± standard error (SE). No statistical methods were used to predetermine the sample size, but our sample sizes are similar to those generally employed in the field. n ≥ 3 mice were used for each experiment and the numbers for animals used for each experiment are stated in the figure legends.

## Acknowledgements

We thank past and present members of the Joyner laboratory for discussions and technical help, especially Dr. Sandra Wilson for initiating studies on the cerebellar nuclei in *En1/2* mutants. We are grateful to Dr. Miguel Torres for providing a polyclonal antibody to all isoforms of mouse MEIS2. We thank Drs. H Saito, N Suzuki and P Faust for the *Pcp2^Cre* mouse line. This work was supported by grants from the NIH to ALJ (NIMH R37MH085726 and NINDS R01NS092096) and RW (NINDS

5F32NS080422), and a National Cancer Institute Cancer Center Support Grant [P30 CA008748-48]. NSB was supported by a NYSTEM postdoctoral fellowship (#C32599GG). ZW was supported by the Kavli Neural Systems Institute at the Rockefeller University.

## Additional information

### Funding

| Funder | Grant reference number | Author |
|---|---|---|
| National Institute of Mental Health | R37MH085726 | Alexandra L Joyner |
| National Institute of Neurological Disorders and Stroke | R01NS092096 | Alexandra L Joyner |
| National Cancer Institute | P30CA008748-48 | Alexandra L Joyner |
| National Institute of Neurological Disorders and Stroke | F32NS080422 | Ryan T Willett |
| New York State Stem Cell Science | C32599GG | N Sumru Bayin |
| Kavli Neural Systems Institute at the Rockefeller University | | Zhuhao Wu |

The funders had no role in study design, data collection and interpretation, or the decision to submit the work for publication.

### Author contributions

Ryan T Willett, Conceptualization, Formal analysis, Funding acquisition, Investigation, Writing—original draft, Writing—review and editing; N Sumru Bayin, Formal analysis, Supervision, Investigation, Writing—review and editing; Andrew S Lee, Anjana Krishnamurthy, Alexandre Wojcinski, Formal analysis, Investigation, Writing—review and editing; Zhimin Lao, Daniel Stephen, Katherine L Dauber-Decker, Investigation, Methodology; Alberto Rosello-Diez, Grant D Orvis, Zhuhao Wu, Resources, Methodology; Marc Tessier-Lavigne, Resources, Funding acquisition; Alexandra L Joyner, Conceptualization, Data curation, Formal analysis, Supervision, Funding acquisition, Writing—original draft, Project administration, Writing—review and editing

### Author ORCIDs

N Sumru Bayin https://orcid.org/0000-0003-4371-855X
Anjana Krishnamurthy https://orcid.org/0000-0002-4584-1722
Zhuhao Wu http://orcid.org/0000-0002-2471-0555
Alexandra L Joyner https://orcid.org/0000-0001-7090-9605

### Ethics

Animal experimentation: This study was performed in strict accordance with the recommendations in the Guide for the Care and Use of Laboratory Animals of the National Institutes of Health. All of the animals were handled according to approved institutional animal care and use committee (IACUC) under an approved protocol from MSKCC (Protocol no: 07-01-001).

### Decision letter and Author response

Decision letter https://doi.org/10.7554/eLife.50617.sa1
Author response https://doi.org/10.7554/eLife.50617.sa2

## Additional files

### Supplementary files

• Transparent reporting form

### Data availability

No datasets were generated.

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
