## [Decision Letter]

**Acceptance summary:**

This study by Willett et al., from the lab of Alexandra Joyner, explains how cerebellum size scales to the cerebellar nuclei (dCN)during their assembly. Using conditional deletion of engrailed genes, expressed in excitatory cerebellar nuclei neurons, the authors investigated how neuron numbers are scaled across the cerebellar circuit. The number of Purkinje cells is matched to that of excitatory cerebellar nuclei (eCN) neurons, through eCN regulation of Purkinje cell survival, and Purkinje cell regulation of granule cell proliferation, via secreted sonic hedgehog. This matching depends on a trophic effect exerted by eCN neurons via Purkinje afferents, with implications for alterations of behavior observed in the mutant animals. Such rules of cell numbers vis a vis targets, and size regulation of gross brain structure, will be of great interest to those studying circuits in other CNS regions.

**Decision letter after peer review:**

[Editors’ note: a previous version of this study was rejected after peer review, but the authors submitted for reconsideration. The first decision letter after peer review is shown below.]

Thank you for submitting your work entitled "Cerebellar nuclei neurons dictate growth of the cortex through developmental scaling of presynaptic Purkinje cells" for consideration by *eLife*. Your article has been reviewed by a Senior Editor, a Reviewing Editor, and three reviewers. The reviewers have opted to remain anonymous.

Our decision has been reached after consultation between the reviewers. Based on these discussions and the individual reviews below, we regret to inform you that your work will not be considered further for publication in *eLife*.

The reviewers found your study on how neuron numbers are scaled across the cerebellar circuit of great interest. You propose that the number of PCs is matched to that of eCN neurons, through cerebellar nuclei neuron regulation of Purkinje cell survival, and Purkinje cell regulation of granule cell proliferation. This matching depends on a trophic effect exerted by eCN neurons via PC afferents, with implications for alterations in these circuits on the behavioral changes observed in the mutant animals.

In our online consultation amongst the reviewers, the reviewers called this a "lovely study", a well conducted and valuable descriptive characterization of the developmental scaling of cerebellar cells. Your proposed principle of matching cell number of afferents and targets could indeed be applicable to other circuits. Nonetheless, even with this appealing overarching hypothesis and clear data presentation consistent with the hypothesis, the reviewers feel that the interesting ideas you presented were not tested, and the data lack any indication of the causal chain of events that would regulate the relatively stable proportions of cell numbers.

Some aspects the reviewers thought were overinterpreted include:

1) Target-dependence of Purkinje neuron survival (concerns of granule cell contribution, and extension to more lateral sites).

2) The matching mechanisms based on target-dependent support does not appear to underpin a physiological depletion of supernumerary PCs (e.g., no evidence for developmental cell death in wt samples), suggesting that PCs might be already matched to eCN at their birth.

3) The generalization that such selection is 'the' developmental mechanism that matches numbers of cerebellar nuclei neurons to that of cortical cells.

These aspects are not simply details but constitute the core hypothesis of the paper.

The study as is presents a set of observations on approximate scaling/compensation within a circuit. Addressing the suggested revisions on these descriptive aspects could be done within two months. However, you would then have to play down the very interesting discussions about scaling, and this approach would lessen the interest of the study. Because testing the model experimentally without confounding components would most likely take much longer, we regret that *eLife* cannot proceed with the manuscript at this time. If you feel you can address these issues some time in the future, we would be happy to consider a new version of the manuscript that addresses the reviewers' concerns.

*Reviewer #1:*

In this study Willett and collaborators investigate the mechanisms of cerebellar development in *Atoh-En1/2* conditional mutants. They nicely show that defined sectors of the mutant vermis display a reduced area and in those sectors both cortical layers and neuron numbers are overall proportionally scaled. Moreover, the number of granule cells and interneurons is reduced to fit the decreased number of PCs. The authors also find that the number of excitatory neurons in mutant medial and intermediate cerebellar nuclei is reduced due to cell death during embryonic development. This associates with PC death, thus explaining the decreased PC number. On these bases, the authors propose that the number of PCs is matched to that of eCN neurons, which would therefore act as the ultimate organizer of cortical morphogenesis. They also suggest that this matching depends on a trophic effect exerted by eCN neurons on their PC afferents.

The presented experiments are well performed, and the results add new data, important to understand cerebellar development. They further propose a mechanism that has general implications for CNS circuit formation.

To fully strengthen their data and clarify their interpretation, the authors should address these points:

- According to the proposed model, and based on the reduction of eCN in the intermediate nucleus, one would predict that the number of PCs is also reduced in the lateral vermis and paravermis, where PCs projecting to this nucleus are located. Nonetheless, while alterations in the vermis of mutants are well documented (and detailed here in mid-sagittal sections), the extent of defects related to variations in PC numbers in those other regions remains unclear. Thus, the assessment of PC scaling also at those other sites is relevant to strengthen the results and the ensuing proposed model.

- The matching mechanisms based on target-dependent support as proposed by the authors does not appear to underpin a physiological depletion of supernumerary PCs (no evidence for developmental cell death in wt samples), suggesting that PCs are already matched to eCN at their birth. This leaves open the possibility that a mitogen-based matching mechanism operates on PC progenitors at earlier developmental stages, similar to postnatal Shh-mediated actions. However, the authors only discuss target-dependence as a scaling mechanism for PC:eCN ratio. Can the authors comment on this?

- It is intriguing to speculate that a specific increase in eCN number in the lateral nuclei underlies the expansion of human cerebellar hemispheres. Is there any evidence from imaging studies or comparative neuroanatomy to support this hypothesis?

Further remarks:

- Figure 1K, M Despite the ratio of interneurons to PCs in *Atoh-En1/2* CKOs does not differ from wt mice, the density of ML interneurons increases in the ASec and CSec at the midline. This result is intriguing: is there any hypotrophy of PC?

- Figure 7. In the pictures (F-I) some labelled PCs appear tagged with distinct colors in seemingly corresponding lobules. However, according to the experimental design, distinct tags should selectively label PCs in different cerebellar areas. Please, clarify this point.

*Reviewer #2:*

This manuscript by Willett et al., examines the interactions among cell types in the developing cerebellum, and the ways in which the neuron numbers are scaled across a circuit. They describe a change in overall cerebellar architecture, with a decrease in size of the central vermis, in a conditional knock out in which the homeobox genes engrailed1 and 2 (En1 and 2) are eliminated in *Atoh1*-expressing neurons. Based on the regional changes in cerebellar nuclei, Purkinje cells and granule cells, they propose a model in which the cerebellar nuclei neurons regulate survival of the Purkinje cell afferents, and the Purkinje cells regulate proliferation of granule cells. They implicate alterations in these circuits in behavioral changes observed in the mutant animals. Overall the studies are well executed and documented, and the potential implications are interesting.

1) A major concern is that there is not strong evidence for the inference that Purkinje cell number is largely dictated by selective survival while cerebellar granule cell number is largely dictated by regulated proliferation. All cell numbers are "set" by regulated proliferation and survival. Clearly there are mutations in which granule cell numbers are altered due to impaired survival rather than altered proliferation. Thus, the data here, while interesting, do not really address a novel question or scaling mechanism.

2) The anatomic basis for the behavioral deficits observed is not well established. Other non-cerebellar cells such as cells within the spinal cord or those in the vestibular system also express *Atoh1*. Are these other areas affected in the CKO and could they contribute to the behavioral changes observed?

3 The ratio of PCs to eCN neurons is 1.4 to 1. This might suggest that other regulators are important, rather than the idea that eCNs are the major determinant of PC cell number, but that the survival promoting mechanism is imprecise.

*Reviewer #3:*

The authors look at changes to the development of the cerebellum when Engrailed (En) 1 and 2 are conditionally knocked out in the *Atoh1* lineage. The mutation leads to a non-uniform shrinkage of the cerebellum. However, the relative proportions principal cerebellar neurons within affected regions of the (medial) cortex remain constant. The densities of granule cells and Purkinje cells are the same. The density of molecular layer interneurons increases (1.2-1.3 fold). There is a 63% loss in number of excitatory neurons (eCN) in the medial and intermediate cerebellar nuclei. The ratio of Purkinje cells to eCN is marginally increased (1.4 fold). Mutants mice show a motor and grip deficit.

A previous report (using transgenic reporters) described En1 and En2 expression in the *Atoh1* lineage (Wilson et al., 2011): It showed that the medial EGL expresses En1 (exclusively) and En2 (predominantly). En1 is expressed in CN (highest laterally) and En2 only in the most medial CN. Here the authors show only relatively weak (Figure 4E) or restricted (Figure 4L,M) En1/2 protein expression in the CN. Nevertheless, there is a very clear targeted cell death in only medial CN from e15.5 (Figure 5) coupled with a lower number of FoxP2-positive cells (Purkinje cell linear: Figure 6). The cell density in the Purkinje layer itself, however, is constant (Figure 2G). Cell death in in the FoxP2 population was not examined, however, axon labelling (Figure 7) suggests that Purkinje cells in the vermis will have predominantly have targeted dying neurons of the medial eCN.

The paper beautifully illustrates the proportional scaling of cell number that characterises cerebellar growth and points towards cell communication as a core feature of this process. The fact that En1/2 expression in Purkinje cells is normal in this model means that changes in cell number at e17.5 must be downstream of changes in the *Atoh1* lineage. The authors propose (from an early stage in the paper) that this is specifically retrograde to eCN loss.

Overall, this paper presents an attractive hypothesis that owes much to classical studies of retrograde regulation of cell death (Cowan, Fawcett, OLeary, 1984), particularly in the visual system, and the neurotrophic hypothesis (Levi-Montalcini and Hamburger), which both need some form of acknowledgement. It is interesting in the context of the massive expansion of the dentate (lateral) nucleus in primates in parallel to the cerebellar hemispheres, and also interesting to consider in the context of a lack of cerebellar nuclei in teleost fish.

To me, there seem to be two main questions.

- Is this the right experiment to test this hypothesis?

- If not an optimal study, do the result unambiguously support the conclusion that is so forcefully expressed in the abstract and introduction?

1) Is this the right experiment?

The optimal approach would be to selectively delete a subset of eCN using genes or combination in a transectional approach to limit mutation to the cerebellum. All effects would then be downstream of the loss of eCN. The authors experimental paradigm goes some way towards this. However, there are confounding factors. They acknowledge that pre-cerebellar En1/2 populations may be affected and a cause of hypoplasia. However, the main cause for concern should be the prominent expression of En1 and to a lesser extent En2 in the medial EGL (Wilson et al., 2011) at e17.5. Could Purkinje cell layer assembly be affected by events in the EGL? There is no synaptic link at this stage, however there is reciprocal signalling both from Purkinje to EGL (via Shh) and EGL to Purkinje cells (via reelin).

2) Do the results support the conclusions?

This is a beautifully documented analysis of scaling within the cerebellum and the changes in cell number, density and overall size are unambiguous. The authors are very keen to assert that there is a cascade of change that stems from eCN death at e15.5. There are a couple of key points that weaken this case.

a) Does eCN cell death occur because of the cell-autonomous loss of En1 and En2? Unclear. The characterisation of expression of En1/2 within the affected CN is quite poor. Unless I have misunderstood, there is only a very small population in the intermediate nucleus that are double labelled for TFTom and EN1/2 (Figure 4L-N). This needs to be better explained.

b) Could a smaller Purkinje cell layer result in an anterograde loss of eCN? Yes. This certainly is the case in the Lurcher mouse (Heckroth, 1994), although the timing of Purkinje cell loss is later that in the model described here.

In both cases the answers are not a strong enough to warrant the forcefulness of the conclusions – the model remains a hypothesis in my view. This means that while the paper shows a beautiful example of scaling and an unexpected motor deficit due to a smaller medial cerebellum, there is no clear discrimination of the causal chain of events that regulates the relatively stable proportions of cell numbers.

[Editors’ note: what now follows is the decision letter after the authors submitted for further consideration.]

Thank you for resubmitting your work entitled "Cerebellar nuclei excitatory neurons regulate developmental scaling of presynaptic Purkinje cell number and organ growth" for further consideration at *eLife*. Your revised article has been favorably evaluated by Marianne Bronner (Senior Editor), a Reviewing Editor, and three reviewers. The manuscript has been improved but there are some remaining issues that need to be addressed before acceptance, as outlined below:

The reviewers, two previous and one new, favor the novel conclusion of your studies that he cerebellum size scales to the early assembled cerebellar nuclei. The results are valid and suitable for publication in *eLife*, and you have appropriately addressed the points raised by the previous reviewers.

However, although none of the criticisms of your resubmission require additional experiments/analyses, all point to issues that should be better discussed and amended. From the reviews, see below, and the reviewer consultation, we agree that the Discussion section is lacking relative to the Results section that include your new data added to the original submission last year.

The reviewers noted a lack of clarity in the conclusions and especially in the Discussion section. There are five areas that need improvement:

1) The new DTA data has almost entirely superseded the use of En conditional mice in support of the main conclusions of the study. The Discussion section should to be overhauled to clearly contrast the outcomes of these two experiments.

As the Discussion section is currently structured, the En-conditional mice seem to introduce confusion:a) They generate a pattern of regional PC loss that is difficult to explain: "Detailed mapping of the eCN that die in each En1/2 CKO and the innervation patterns of PCs to spatial regions of each CN will be required to fully understand the phenotypes".b) They generate behavioral deficits that also have an unclear origin, currently attributed to "the imprecise proportions of eCN and PCs" or a smaller cerebellum being unable to "support all the long range neural circuits needed for full motor function".

2) The information and statements in the rebuttal letter and the spirit of the changes in the text do not quite match up.

For example, on behavior, the relatively clear rebuttal statement that "some of the behavior deficits could therefore be due to defects in cell types that are not the focus of the paper" is not represented in the Discussion (some anatomical "phenotypes" could be "secondary (cell-nonautonomous) effect of loss of En1/2 or killing with DT of neurons outside the cerebellum".

3) The comment of reviewer 2 on additional mechanisms possibly participating in the scaling (e.g. precursor proliferation) already emerged in the former round of revision and was recognized in your rebuttal. We suggest that you tone down statements presenting the observed scaling back as 'the" mechanism that determines the cell number of their pre-synaptic partner PCs.

4). Clarity is need on the possible mechanistic explanation for the regional variability you see, namely, the mechanisms of cell death. As reviewer 2 notes, "'Growth' can reflect many different processes. Is this only due to apoptosis of neurons generated? Is there an effect on progenitor proliferation?" In addition, explanation is needed on regional differences in cerebellar growth and in the Methods, how the ratio of PC to eCN in the medial nucleus is calculated.

5) The reviewers also call for more contextual information about the potential role of neurotrophins, especially citation of classical literature on neurotrophins.

*Reviewer #1:*

The manuscript has been extensively revised since its last submission with the addition of new data. These have addressed comments arising from the previous submission and the work provides convincing support for the hypothesis that the cerebellum size scales to the early assembled cerebellar nuclei.

*Reviewer #2:*

In this manuscript, Willet et al., investigate the effect of deleting Engrailed1/2 specifically from the eCN neuronal population in the cerebellum, demonstrating that when eCN neurons are lost due to the deletion of EN1/2, there is a concomitant decrease in neuron number across the entire circuit, with reduced numbers of Purkinje cells and granule cells. When they delete En1/2 from the cerebellar nuclei, there is a reduction of neurons in the medial and intermediate, but not lateral, nuclei. The authors beautifully demonstrate that there is a corresponding loss of the Purkinje cell presynaptic partners, and discuss the scaling of neuron numbers to maintain proportionality in the circuitry.

There are many interesting and novel aspects to this study. Using developmental regulation of the timing of the *Atoh1-Cre* to remove EN1/2 specifically from the CNs or the GPCs highlights the different EN functions in different neuronal populations. Tracing the projections of specific PC populations to their respective CN regions to show that the PCs that project to the medial and intermediate eCN are the ones that are lost. The use of DTA to kill eCN confirmed that the rest of the cerebellar populations scale down when eCN neurons are decreased, although this decrease occurred in the cerebellar hemispheres as well as the vermis and paravermis.

However, there are a number of questions associated with this study that are not addressed, and could at least be raised in the Discussion section.

1) They mention (subsection “*En1/2* are required in the eCN but not GCPs for growth of the cerebellar cortex”) that En1/2 are necessary in the eCN compartment to promote cerebellar growth, but "growth" can reflect many different processes. Is this only due to apoptosis of neurons generated? Is there an effect on progenitor proliferation?

2) They demonstrate that PCs are generated in normal numbers and there is increased cell death, suggesting that they lack trophic support from their target eCN neurons, although the PCs did not scale down completely. However, why there are fewer eCN neurons in the *eCN-En1/2* cKO is not addressed, what is the mechanism by which the loss of En1/2 leads to the death of these eCN neurons?

3) There is no explanation or hypothesis provided in the Discussion for the regional differences in cerebellar growth in the *Atoh1-En1/2* CKO, that the decreased size is only seen in the vermis and paravermis and not in the hemispheres, and only in the anterior and central, but not posterior sector. Is En1/2 equally expressed in all these regions? Does the *Atoh1-Cre* remove En1/2 from those regions that don't show decreased size after they have already developed?

4) There was no scaling of the PV-positive inhibitory interneurons (Figure 5), possible reasons for that could be discussed.

*Reviewer #3:*

The authors have addressed all concerns previously expressed. With a new set of elegant genetic experiments, they have now significantly strengthened the evidence for:

- a developmental dependency of PCs on eCNs for they survival – this applies to vermis, paravermis and hemispheres;

- normal neurogenesis for eCNs and PCs in the examined mutants, and scaling of PC numbers after the beginning of eCN death;

- lack of impact of subtle alterations in granule cell development on the observed PC loss and major cerebellar cortical growth defects.

Thus, taken together, the current data nicely support the view that PC numbers are matched to their postsynaptic partners and show that the cerebellar cortex is then organized proportionally to the PC numbers.

To this reviewer, however, the author's statement at the beginning of the discussion somehow appears to go beyond these conclusions: '.. whereby the earliest-born neurons, the eCN, determine the cell number of their pre-synaptic partner PCs'. According to what the authors write in the rebuttal 'our paper is addressing a.. scaling mechanism, which might be most prevalent after injury or used normally for making minor adjustments'. Here the authors appear to recognize that additional mechanisms should be in place to match eCN and PCs, likely acting during neurogenesis. Therefore, the above reported statement sounds too strong and should be tuned down to better fit the results of this study.

This reviewer is also confused now about how the ratio of PC to eCN in the medial nucleus is calculated. When reading the methods, it seems that the average number of PCs at the vermis midline (e.g. about 350+190 in WT mice, Figure 1L) is divided by the number of neurons in the medial nucleus (about 1000 in corresponding WT mice, Figure 1Q). The result here cannot be 20 as plotted in Figure 1—figure supplement 1M. Have I missed some information? Please, clarify.

[Editors’ note: what now follows is the decision letter after the authors submitted for further consideration.]

Thank you for resubmitting your work entitled "Cerebellar nuclei excitatory neurons regulate developmental scaling of presynaptic Purkinje cell number and organ growth" for further consideration by *eLife*. Your revised article has been evaluated by Marianne Bronner (Senior Editor) and a Reviewing Editor.

We are pleased to inform you that the reviewers were satisfied with your revisions, most of which made your arguments better organized and the text clearer. Reviewer 1, however, had a few additional suggestions for further clarifying your writing, as follows:

Introduction: "In turn, the PC-derived mitogenic factor, SHH, stimulates the proportional proliferation of GC, interneuron and glial progenitors and thus the level of production of all postnatally derived neurons in the cerebellar cortex (Figure 9)"

The suggested change is: rather than "in turn", it would be better to say "Previous research indicates that". This avoids any implication that these insights on Shh arise from this study. In addition, in many places, you cite the influence of Shh without making reference to the previous studies and so this also needs to be amended. Also, Shh is sometimes expressed as SHH – make consistent.

Discussion section: "…as it might not be able to support all the long range neural circuits needed for full motor function."

This hypothesis needs a little elaboration. Is there any evidence from other studies that a small cerebellum, per se, results in motor deficits (i.e., in Shh mutants)?

Discussion section: "One idea for the higher ratios of vermal PCs in lobules 1-8 to eCN in the medial CN in the mutants is that the difference reflects the mechanism by which scaling is attained. In the case of eCN:PC scaling, we propose the process depends on the availability of a survival factor (such as a neurotrophin), since the PCs die after E15.5 when all the neurons have been born, whereas the PC:GC or PC:interneuron scaling depends on the amount of mitogen produced (SHH) which stimulates proliferation of two progenitor pools. One additional possibility related to eCN:PC scaling is that since each PC projects to many eCN (Person and Raman, 2011), if the survival of PCs is dependent on a factor secreted by eCN, then it might be that a greater proportion of PCs can survive than in normal homeostasis because the remaining PCs project to more of the remaining eCN in mutants than normal and thus receive sufficient survival factor."

These two sentences are rather tortuous and difficult to extract the argument. Please consider clarifying. Reviewer 1 tried to re-write the paragraph in a way that was clearer, in this way:

"The higher than expected ratio of PCs in the vermis might reflect the way in which scaling is established. GCs and interneuron numbers depend on a precise and deterministic proliferative response downstream of PC number. However, PC number depends on potentially stochastic retrograde survival factor from the eCN. Moreover, PC survival may be complicated by the presence of collateral axon projections from some PC (Person and Raman, 2011) to eCN less affected by the mutation."

---

## [Author Response]

[Editors’ note: the author responses to the first round of peer review follow.]

[…] Some aspects the reviewers thought were overinterpreted include:1) Target-dependence of Purkinje neuron survival (concerns of granule cell contribution, and extension to more lateral sites).2) The matching mechanisms based on target-dependent support does not appear to underpin a physiological depletion of supernumerary PCs (e.g., no evidence for developmental cell death in wt samples), suggesting that PCs might be already matched to eCN at their birth.3) The generalization that such selection is 'the' developmental mechanism that matches numbers of cerebellar nuclei neurons to that of cortical cells.These aspects are not simply details but constitute the core hypothesis of the paper.The study as is presents a set of observations on approximate scaling/compensation within a circuit. Addressing the suggested revisions on these descriptive aspects could be done within two months. However, you would then have to play down the very interesting discussions about scaling, and this approach would lessen the interest of the study. Because testing the model experimentally without confounding components would most likely take much longer, we regret that eLife cannot proceed with the manuscript at this time. If you feel you can address these issues some time in the future, we would be happy to consider a new version of the manuscript that addresses the reviewers' concerns.

We thank the reviewers for their constructive comments. We agree that additional evidence for a specific role of excitatory cerebellar nuclei neurons (eCN) in scaling the number of Purkinje cell (PCs) would strengthen our hypothesis. We have therefore performed an extensive series of additional genetic experiments that we think should satisfy the reviewer’s concerns and provide direct evidence for our proposed cell number scaling mechanism. We have used a transgenic mouse we made that expresses tTA from *Atoh1* regulatory sequences in the rhombic lip lineage in conjunction with a *tetO-Cre* line and our *En1/2* conditional alleles in order to delete *En1/2* only in embryonic eCN or GCPs, by feeding the mice doxycycline before or after the eCN are generated (E13.5). We find that indeed when *En1/2* are removed from GCPs, there is no decrease in the growth of the cerebellar cortex, but when *En1/2* are removed from the eCN or eCN+GCPs there is a significant decrease in eCN and PC numbers and the cerebellum is smaller than normal. As a second approach to test that PC numbers scale to eCN numbers, we used the same *Atoh1-tTA* in combination with *Atoh1-Cre* in an intersectional approach with a new transgene we made to transiently express attenuated diphtheria toxin (DTA) in eCN at embryonic day ~9-13 (*eCN-DTA*). We indeed found that killing ~40% of eCN results in a reduction in the number of PCs, and reduced growth of the cerebellar cortex. As further evidence that PCs scale to their eCN postsynaptic partners, the eCN that die in *En1/2* CKOs and *eCN-DTA* animals are different, with only the medial and intermediate eCN being reduced in *En1/2* CKOs whereas there is uniform eCN death across the medio-lateral axis in *eCN-DTA* mice. As a consequence, cerebellar size is only reduced in the vermis and paravermis in the *En1/2* CKOs, whereas in *eCN-DTA* the hemispheres were also smaller. Finally, we also used our mosaic mutant analysis approach to identify any minor role of *En1/2* in GCPs by deleting the genes in scattered GCPs at P2 and analyzing the output of GCs at P8. We found there is a minor but significant decrease in the proportion of GCs generated of all the mutant cell (GCPs+GCs) and a complementary increase in the proportion of mutant GCPs, revealing that *En1/2* normally play a minor role in promoting GC differentiation. When all GCPs are mutant for *En1/2* using our GCP-specific approach, there is no measurable long term defect in overall cerebellar growth and the density of the GCs in the internal granule cell layer (IGL) is maintained, but the area of the IGL relative to the cerebellum is slightly reduced. We have expanded our Discussion section and hypothesis to include all our new results.

Reviewer #1:[…] To fully strengthen their data and clarify their interpretation, the authors should address these points:- According to the proposed model, and based on the reduction of eCN in the intermediate nucleus, one would predict that the number of PCs is also reduced in the lateral vermis and paravermis, where PCs projecting to this nucleus are located. Nonetheless, while alterations in the vermis of mutants are well documented (and detailed here in mid-sagittal sections), the extent of defects related to variations in PC numbers in those other regions remains unclear. Thus, the assessment of PC scaling also at those other sites is relevant to strengthen the results and the ensuing proposed model.

As suggested by the reviewer, we have quantified the area and PC numbers in the paravermis and hemispheres of *Atoh1-En1/2* CKO mutants as well as our new models and find a significant decrease in the area and number of PCs in the paravermis but not the hemispheres in all *En1/2* CKOs involving the eCN, and also in the hemispheres in *eCN-DTA* mutants. Since our new embryonic eCN killing model (*eCN-DTA*) and not the *En1/2* CKOs leads to uniform eCN loss across the medio-lateral access, the specific reduction in the area of the hemispheres in the *eCN-DTA* mice provides additional evidence that PC numbers (and growth of the cerebellar) cortex is influenced by the number of their postsynaptic eCN partners.

- The matching mechanisms based on target-dependent support as proposed by the authors does not appear to underpin a physiological depletion of supernumerary PCs (no evidence for developmental cell death in wt samples), suggesting that PCs are already matched to eCN at their birth. This leaves open the possibility that a mitogen-based matching mechanism operates on PC progenitors at earlier developmental stages, similar to postnatal Shh-mediated actions. However, the authors only discuss target-dependence as a scaling mechanism for PC:eCN ratio. Can the authors comment on this?

The reviewer brings up an interesting alternative that we had considered and may well operate during neurogenesis. However, our paper is addressing a later, or second scaling mechanism, which might be most prevalent after injury or used normally for making minor adjustments. As shown in our original manuscript, we found that the number of eCN and PCs are not significantly reduced at E15 in *Atoh1-En1/2* CKOs, and then both are reduced at E17.5. In our new *eCN-En1/2* CKOs, eCN numbers are only partially reduced at E17.5 and PC numbers are normal at E17.5 but both are significantly reduced at P30. Similarly, in our *eCN-DTA* model where eCN are killed before E15.5, PC numbers are not reduced at E15.5 but are reduced at P1 (not shown) and in the adult cerebellum. Thus, PC numbers are not scaled back after the eCN die. Since the number of PCs is initially normal and becomes reduced after the eCN are reduced, we propose a target derived factor for this aspect of number scaling we are studying.

- It is intriguing to speculate that a specific increase in eCN number in the lateral nuclei underlies the expansion of human cerebellar hemispheres. Is there any evidence from imaging studies or comparative neuroanatomy to support this hypothesis?

We agree with the reviewer’s hypothesis and have added the idea to the Discussion section. From our reading of the scant literature on quantification of eCN in humans, there are indeed more in the lateral nuclei, as they are much larger than in mouse and divided into several subgroups, further supporting the idea that the number of eCN dictates the growth of the local cerebellar circuitry.

- Figure 1K, M Despite the ratio of interneurons to PCs in Atho-En1/2 CKOs does not differ from wt mice, the density of ML interneurons increases in the ASec and CSec at the midline. This result is intriguing: is there any hypotrophy of PC?

We have quantified PC size in the *Atoh1-En1/2* CKO mutants and there is no hypertrophy in PCs, see Author response image 1. We have therefore not added the results to the paper.

**Author response image 1. respfig1:** Quantification of PC soma size in *Atoh1-En1/2* CKOs. No overall signficiant difference between mutant and control PCs (One-way ANOVA, p>0.1).

- Figure 7. In the pictures (F-I) some labelled PCs appear tagged with distinct colors in seemingly corresponding lobules. However, according to the experimental design, distinct tags should selectively label PCs in different cerebellar areas. Please, clarify this point.

We apologize for the confusion. The colors were pseudo-colored, as only one viral construct was used and were not meant to correspond to former 7D,E (now Figure 3D,E). We have clarified this in the figure legend.

Reviewer #2:[…] 1) A major concern is that there is not strong evidence for the inference that Purkinje cell number is largely dictated by selective survival while cerebellar granule cell number is largely dictated by regulated proliferation. All cell numbers are "set" by regulated proliferation and survival. Clearly there are mutations in which granule cell numbers are altered due to impaired survival rather than altered proliferation. Thus, the data here, while interesting, do not really address a novel question or scaling mechanism.

As described above, we would like to note that the PCs and eCN are born in normal numbers in *Atoh1-En1/2* CKOs, and then they are both scaled back. This is a phenotype not previously described, and we think that with the extensive new genetic studies (see above) our work provides novel insights into a second type of scaling.

2) The anatomic basis for the behavioral deficits observed is not well established. Other non-cerebellar cells such as cells within the spinal cord or those in the vestibular system also express Atoh1. Are these other areas affected in the CKO and could they contribute to the behavioral changes observed?

The reviewer brings up a good point that we had previously mentioned in the Discussion section. We have previously shown that *En1/2* are not expressed in spinal cord *Atoh1*-derived cells (Sillitoe, 2010). *En1/2* do have a dynamic and complicated expression pattern in some precerebellar nuclei derived from the *Atoh1*-lineage. Some of the behavior deficits could therefore be due to defects in cell types that are not the focus of the paper. Unfortunately, all the available Cre lines that are specific to the eCN are also expressed in the precerebellar nuclei, including our new tTA/Cre approach. We nevertheless think the behavior results are worthwhile including in the paper, but we are open to removing them given all the additional experimental results that have been added.

3 The ratio of PCs to eCN neurons is 1.4 to 1. This might suggest that other regulators are important, rather than the idea that eCNs are the major determinant of PC cell number, but that the survival promoting mechanism is imprecise.

We agree with the reviewer that since it appears that PCs do not fully scale back to the reduced number of eCN, now seen in additional models, the scaling mechanism cannot be as simple as we previously stated. We have elaborated on this idea in the Discussion section.

Reviewer #3:[…] To me, there seem to be two main questions.- Is this the right experiment to test this hypothesis?- If not an optimal study, do the result unambiguously support the conclusion that is so forcefully expressed in the abstract and introduction?1) Is this the right experiment?The optimal approach would be to selectively delete a subset of eCN using genes or combination in a transectional approach to limit mutation to the cerebellum. All effects would then be downstream of the loss of eCN. The authors experimental paradigm goes some way towards this. However, there are confounding factors. They acknowledge that pre-cerebellar En1/2 populations may be affected and a cause of hypoplasia. However, the main cause for concern should be the prominent expression of En1 and to a lesser extent En2 in the medial EGL (Wilson et al., 2011) at e17.5. Could Purkinje cell layer assembly be affected by events in the EGL? There is no synaptic link at this stage, however there is reciprocal signalling both from Purkinje to EGL (via Shh) and EGL to Purkinje cells (via reelin).

We agree with the reviewer that the paper would be strengthened by having CKOs that remove *En1/2* only in the eCN or GCPs in the embryonic cerebellum. As described above, we have now performed multiple complicated genetic experiments to dissect the roles of *En1/2* in eCN and GCPs during cerebellar development. As described above, we made a transgenic mouse that expresses tTA from *Atoh1* regulatory sequences in the rhombic lip lineage and combined it with a *tetO-Cre* line and our *En1/2* conditional alleles and fed the mice doxycycline after or before the eCN are generated (E13.5) in order to delete *En1/2* only in the eCN or GCPs in the embryonic cerebellum. We find that indeed when *En1/2* are removed from GCPs, there is no decrease in the growth of the cerebellar cortex, but when *En1/2* are removed from the eCN there is a significant decrease in eCN and PC numbers and the cerebellum is smaller than normal. There are subtle differences in the regional hypoplasia using the two strategies, which likely reflects either the subgroup of eCN that Cre is expressed in using *Atoh1-Cre* or *Atoh1-tTA* or more likely that the timing of Cre since Cre will be slightly delayed using tTA. We also used our mosaic mutant analysis approach to identify any minor role of *En1/2* in GCPs by deleting the genes in scattered GCPs at P2 and analyzing the output of GCs at P8. We found there is a minor but significant decrease in the proportion of GCs generated of all the mutant cell (GCPs+GCs) and a complementary increase in the proportion of mutant GCPs, revealing that *En1/2* normally promote GC differentiation. When all GCPs (and not eCN) are mutant for *En1/2* using *Atoh1-tTA*, there is no long term defect in cerebellar growth or the density of the GCs in the internal granule cell layer (IGL), but the area of the IGL as a proportion of cerebellar area is slightly reduced. We have expanded our Discussion section to include all our new results.

2) Do the results support the conclusions?This is a beautifully documented analysis of scaling within the cerebellum and the changes in cell number, density and overall size are unambiguous. The authors are very keen to assert that there is a cascade of change that stems from eCN death at e15.5. There are a couple of key points that weaken this case.a) Does eCN cell death occur because of the cell-autonomous loss of En1 and En2? Unclear. The characterisation of expression of En1/2 within the affected CN is quite poor. Unless I have misunderstood, there is only a very small population in the intermediate nucleus that are double labelled for TFTom and EN1/2 (Figure 4L-N). This needs to be better explained.

We have expanded our analysis of EN1/2 expression in the embryonic eCN, including using a new *En2* HAtagged allele we generated. *En1/2* expression is dynamic and most broadly in the eCN before E15.5, and then remains in a subset of eCN. There is growing evidence from published results and our current study that EN1/2 play a transient role in promoting differentiation and survival of several neuron types. In the cells that maintain *En1/2* expression, we speculate that the genes play an additional role in neuron function. As we have now deleted *En1/2* only in the eCN and still find cell death in the TdTom+ eCN, we have shown that EN1/2 are important for cell survival.

b) Could a smaller Purkinje cell layer result in an anterograde loss of eCN? Yes. This certainly is the case in the Lurcher mouse (Heckroth, 1994), although the timing of Purkinje cell loss is later that in the model described here.

We think this is unlikely in our mutants, since in the published papers death of CN neurons in Lurcher mutants occurs months after the loss of PCs. Since earlier killing leads to rapid replacement of PCs, we cannot kill the PCs at an earlier stage.

In both cases the answers are not a strong enough to warrant the forcefulness of the conclusions – the model remains a hypothesis in my view. This means that while the paper shows a beautiful example of scaling and an unexpected motor deficit due to a smaller medial cerebellum, there is no clear discrimination of the causal chain of events that regulates the relatively stable proportions of cell numbers.

We think that our new experimental studies, including specific DTA-mediated embryonic killing of 40% of the eCN throughout the medial-lateral axis, add substantially to our paper and provide strong support for our hypothesis.

[Editors' note: the author responses to the re-review follow.]

The reviewers, two previous and one new, favor the novel conclusion of your studies that he cerebellum size scales to the early assembled cerebellar nuclei. The results are valid and suitable for publication in eLife, and you have appropriately addressed the points raised by the previous reviewers.However, although none of the criticisms of your resubmission require additional experiments/analyses, all point to issues that should be better discussed and amended. From the reviews, see below, and the reviewer consultation, we agree that the Discussion section is lacking relative to the Results section that include your new data added to the original submission last year.The reviewers noted a lack of clarity in the conclusions and especially in the Discussion section. There are five areas that need improvement:1) The new DTA data has almost entirely superseded the use of En conditional mice in support of the main conclusions of the study. The Discussion section should to be overhauled to clearly contrast the outcomes of these two experiments.

We had previously done this where it seems appropriate in the Discussion section but have now added additional reference to both types of mutants.

As the Discussion section is currently structured, the En-conditional mice seem to introduce confusion:a) They generate a pattern of regional PC loss that is difficult to explain: "Detailed mapping of the eCN that die in each En1/2 CKO and the innervation patterns of PCs to spatial regions of each CN will be required to fully understand the phenotypes".b) They generate behavioral deficits that also have an unclear origin, currently attributed to "the imprecise proportions of eCN and PCs" or a smaller cerebellum being unable to "support all the long range neural circuits needed for full motor function".

We have addressed all the specific comments of the reviewers below and tried to clarify the conclusion or hypotheses we made in the Discussion section. (a) Data presented in Figure 3 shows that the central region where the lobule size and PC loss is the greatest in *Atoh1-En1/2* CKOs project to the area of the medial eCN (posterior) where the greatest eCN loss is observed (Figure 1Q and Figure 1—figure supplement 3). (b) Interpretation of the behavioral phenotypes is discussed further below in the specific comments of the reviewers including that it could involve hindbrain defects.

2) The information and statements in the rebuttal letter and the spirit of the changes in the text do not quite match up.For example, on behavior, the relatively clear rebuttal statement that "some of the behavior deficits could therefore be due to defects in cell types that are not the focus of the paper" is not represented in the Discussion (some anatomical "phenotypes" could be "secondary (cell-nonautonomous) effect of loss of En1/2 or killing with DT of neurons outside the cerebellum".

We have clarified that by ‘phenotype’ we mean both cellular defects and behavior deficits.

3) The comment of reviewer 2 on additional mechanisms possibly participating in the scaling (e.g. precursor proliferation) already emerged in the former round of revision and was recognized in your rebuttal. We suggest that you tone down statements presenting the observed scaling back as 'the" mechanism that determines the cell number of their pre-synaptic partner PCs.

We did not mean to imply that this is the only mechanism, certainly proliferation of VZ progenitors and nestinexpressing progenitors after birth are critical step. We have tried to clarify and tone down our conclusions throughout the paper.

4). Clarity is need on the possible mechanistic explanation for the regional variability you see, namely, the mechanisms of cell death. As reviewer 2 notes, "'Growth' can reflect many different processes. Is this only due to apoptosis of neurons generated? Is there an effect on progenitor proliferation?" In addition, explanation is needed on regional differences in cerebellar growth and in the Methods, how the ratio of PC to eCN in the medial nucleus is calculated.

We addressed the specific comments of reviewer 2 on growth and regional differences below and modified our conclusions accordingly.

How the PC to eCN ratio was calculated was previously described in the Materials and ethods section of our resubmission. We have expanded the material and methods section to add more details based on the reviewers’ comments. In summary, PC to eCN ratio was obtained by dividing the extrapolated number of PCs in the ASec and the CSec of the half vermis (average PC number in the ASec and CSec of a midline section multiplied by half the number of 14 μm-thick sections in half a CB) divided by the number of medial eCN present on every second section of half a CB using our two different methods (semi-automated eCN counting of cell type specific En1/2 CKOs and eCN-DTA mutants or stereology for *Atoh1-En1/2* cKOs).

5) The reviewers also call for more contextual information about the potential role of neurotrophins, especially citation of classical literature on neurotrophins.

We had toned down our discussion of the neurotrophic theory in response to the previous comments that we do not have direct evidence for a particular neurotrophin, and at the same time we added the classical papers requested in our resubmission (Levi-Montalcini and Hamburger, 1951; Cowan et al., 1984). We have now added back an introduction to the neurotrophic theory in the Introduction and mention it in the Discussion section.

Reviewer #1:The manuscript has been extensively revised since its last submission with the addition of new data. These have addressed comments arising from the previous submission and the work provides convincing support for the hypothesis that the cerebellum size scales to the early assembled cerebellar nuclei.Reviewer #2:[…] However, there are a number of questions associated with this study that are not addressed, and could at least be raised in the Discussion section.1) They mention (subsection “En1/2 are required in the eCN but not GCPs for growth of the cerebellar cortex”) that En1/2 are necessary in the eCN compartment to promote cerebellar growth, but "growth" can reflect many different processes. Is this only due to apoptosis of neurons generated? Is there an effect on progenitor proliferation?

We have edited the sentences as follows:

“Taken together, our data demonstrate that En1/2 are necessary in the eCN compartment of the rhombic liplineage to promote postnatal growth of the cerebellar cortex. The numbers of interneurons and granule cells in the cerebellar cortex are scaled proportionately to the number of PCs, likely through decreased SHHstimulated proliferation of progenitors in *eCN-En1/2* CKOs as in *Atoh1-En1/2* CKO.”

2) They demonstrate that PCs are generated in normal numbers and there is increased cell death, suggesting that they lack trophic support from their target eCN neurons, although the PCs did not scale down completely. However, why there are fewer eCN neurons in the eCN-En1/2 cKO is not addressed, what is the mechanism by which the loss of En1/2 leads to the death of these eCN neurons?

Determining the mechanism for the death of the eCN in *eCN-En1/2* CKOs (and *eCN+GCP-En1/2* CKOs) is beyond the scope of the paper. In Figure 7 and Figure 7—figure supplement 2 we show that eCN are born in normal numbers and then die. The reason they are born normally could be because EN1/2 protein is not lost in the mutants until after they are born. Why EN1/2 are required for survival of only a subset of eCN could be because they are only expressed in a subset of eCN after E14 and/or because they have different functions in different eCN.

3) There is no explanation or hypothesis provided in the Discussion for the regional differences in cerebellar growth in the Atoh1-En1/2 CKO, that the decreased size is only seen in the vermis and paravermis and not in the hemispheres, and only in the anterior and central, but not posterior sector. Is En1/2 equally expressed in all these regions? Does the Atoh1-Cre remove En1/2 from those regions that don't show decreased size after they have already developed?

We are sorry that we had not explained this clearly enough in the text. We had tried to explain that the reason for the decrease in size only in the vermis and paravermis and not in the hemispheres is because the eCN are only reduced in the medial and intermediate CN and these are the CN that the Purkinje cells in the vermis and paravermis project – the hemispheres project to the lateral CN. In terms of the anterior and central sectors, but not posterior sector, as we had stated the posterior sector Purkinje cells do not project to the CB but instead to the vestibular nucleus (Results section). We have tried to modify our explanations.

4) There was no scaling of the PV-positive inhibitory interneurons (figure 5), possible reasons for that could be discussed.

This is not correct. We show that the density of PV+ neurons is normal and the ML is decreased, thus the number of PV+ neurons is decreased in *eCN-En1/2* and *eCN+GCP-En1/2* CKOs.

Reviewer #3:[…] To this reviewer, however, the author's statement at the beginning of the discussion somehow appears to go beyond these conclusions: '.. whereby the earliest-born neurons, the eCN, determine the cell number of their pre-synaptic partner PCs'. According to what the authors write in the rebuttal 'our paper is addressing a.. scaling mechanism, which might be most prevalent after injury or used normally for making minor adjustments'. Here the authors appear to recognize that additional mechanisms should be in place to match eCN and PCs, likely acting during neurogenesis. Therefore, the above reported statement sounds too strong and should be tuned down to better fit the results of this study.

We have revised the beginning of the Discussion section as follows:

In summary, the results of our studies show that development of the cerebellum involves a sequence of events that includes the earliest-born neurons, the eCN, having an influence on the final number of their pre-synaptic partner PCs by supporting their survival after PC neurogenesis is complete. In turn, the PCderived mitogenic factor, SHH, stimulates the proportional proliferation of GC, interneuron and glial progenitors and thus the level of production of all postnatally derived neurons in the cerebellar cortex (Figure 9).

This reviewer is also confused now about how the ratio of PC to eCN in the medial nucleus is calculated. When reading the methods, it seems that the average number of PCs at the vermis midline (e.g. about 350+190 in WT mice, Figure 1L) is divided by the number of neurons in the medial nucleus (about 1000 in corresponding WT mice, Figure 1Q). The result here cannot be 20 as plotted in Figure 1—figure supplement 1M. Have I missed some information? Please, clarify.

We apologize for the confusion. As stated above in more detail, the number of PCs used for this calculation is an estimate of the total number of PC in the ASec and CSec of a half vermis not the number of PCs in the ASec + CSec of one midline section. We understand that this is a crude estimate, but it was used for purposes of comparing controls to mutants. We have revised the Materials and methods section to explain our calculation in more detail.

[Editors' note: the author responses to the re-review follow.]

We are pleased to inform you that the reviewers were satisfied with your revisions, most of which made your arguments better organized and the text clearer. Reviewer 1, however, had a few additional suggestions for further clarifying your writing, as follows:Introduction"In turn, the PC-derived mitogenic factor, SHH, stimulates the proportional proliferation of GC, interneuron and glial progenitors and thus the level of production of all postnatally derived neurons in the cerebellar cortex (Figure 9)"The suggested change is: rather than "in turn", it would be better to say "Previous research indicates that". This avoids any implication that these insights on Shh arise from this study. In addition, in many places, you cite the influence of Shh without making reference to the previous studies and so this also needs to be amended. Also, Shh is sometimes expressed as SHH – make consistent.

As requested, we have replaced "In turn,” with "Previous research indicates that". We certainly were not trying to imply we discovered the role of SHH in stimulating GCPs and NEPs in this paper, and hence we had added the appropriate references at the end of the sentence (some of which are from our previous work).

I would like to make several comments.

First, this sentence is in the Discussion section not the Introduction.

Second, we had the following references in this sentence, after “(Figure 9)”

“(Corrales et al., 2006; Corrales et al., 2004; De Luca et al., 2015; Fleming et al., 2013; Lewis et al., 2004; Parmigiani et al., 2015; Wojcinski et al., 2017)”

Third, we have added the following references every place we mentioned SHH and had not included any references.

(Corrales et al., 2006; De Luca et al., 2015; Fleming et al., 2013; Lewis et al., 2004; Wojcinski et al., 2017).

Fourth, we are following the accepted nomenclature for mouse genes and proteins. http://www.informatics.jax.org/mgihome/nomen/. We use italics and lower case for genes and RNA (*Shh*) and all capitals for proteins (SHH). We have therefore not changed the nomenclature.

Discussion section: "…as it might not be able to support all the long range neural circuits needed for full motor function."This hypothesis needs a little elaboration. Is there any evidence from other studies that a small cerebellum, per se, results in motor deficits (i.e., in Shh mutants)?

In the subsection “*Atoh1-En1/2* mutants have motor deficits” we had previously included the following sentence first sentence “We next tested whether a smaller cerebellum that has a well scaled cortex can support normal motor behavior, since we had found that irradiated mice with an ~20% reduction in the area of mid-sagittal sections but well scaled layers have no obvious motor defects (Wojcinski et al., 2017).” We are not aware of any other paper that has analyzed behavior in mutants with a normal cyotoarchitecture but a smaller cerebellum. In *Shh* and *Gli* conditional mutants the granule cells and interneurons are not scaled to the number of Purkinje cells, as the Purkinje cells remain as a multilayer and the IGL is greatly diminished.

Based on observations of mice in their cages, we think the mutants have motor defects.

We have added that our mutants have an ~30% reduction in cerebellar size “smaller size of the cerebellum (~30% reduction)” and added the following sentence to the Discussion section:

“In a previous study an ~20% reduction in the cerebellum without disruption of the cytoarchitecture did not lead to obvious motor defects, however the eCN and other cell types were not quantified and the reduction in size was less (Wojcinski et al., 2017).”

Discussion section: "One idea for the higher ratios of vermal PCs in lobules 1-8 […] sufficient survival factor."These two sentences are rather tortuous and difficult to extract the argument. Please consider clarifying. Reviewer 1 tried to re-write the paragraph in a way that was clearer, in this way:"The higher than expected ratio […] less affected by the mutation."

We have replaced the difficult two sentences with the:

“The higher than expected ratio of PCs in the vermis to the medial eCN might reflect the way in which scaling is established. GC and interneuron numbers depend on a precise and deterministic proliferative response to SHH downstream of PC number (Corrales et al., 2006; De Luca et al., 2015; Fleming et al., 2013; Lewis et al., 2004; Wojcinski et al., 2017). In contrast, we propose that PC number depends on stochastic retrograde survival factors from the eCN. Moreover, PC survival could be complicated by the convergence ratio of ~40 PCs to each eCN (Person and Raman, 2011).”